# Convergence of Score-Based Discrete Diffusion Models: A Discrete-Time Analysis

**Zikun Zhang**
School of Mathematical Sciences
Fudan University
zkzhang21@m.fudan.edu.cn

**Zixiang Chen**
Department of Computer Science
University of California, Los Angeles
chenzx19@cs.ucla.edu

**Quanquan Gu**
Department of Computer Science
University of California, Los Angeles
qgu@cs.ucla.edu

## Abstract

Diffusion models have achieved great success in generating high-dimensional samples across various applications. While the theoretical guarantees for continuous-state diffusion models have been extensively studied, the convergence analysis of the discrete-state counterparts remains under-explored. In this paper, we study the theoretical aspects of score-based discrete diffusion models under the Continuous Time Markov Chain (CTMC) framework. We introduce a discrete-time sampling algorithm in the general state space $[S]^d$ that utilizes score estimators at predefined time points. We derive convergence bounds for the Kullback-Leibler (KL) divergence and total variation (TV) distance between the generated sample distribution and the data distribution, considering both scenarios with and without early stopping under reasonable assumptions. Notably, our KL divergence bounds are nearly linear in the dimension $d$, aligning with state-of-the-art results for diffusion models. Our convergence analysis employs a Girsanov-based method and establishes key properties of the discrete score function, which are essential for characterizing the discrete-time sampling process.

## 1 Introduction

Diffusion models (Sohl-Dickstein et al., 2015; Song & Ermon, 2019; Ho et al., 2020) have achieved great empirical success in various generative tasks including audio (Kong et al., 2021; Schneider, 2023), video (Ho et al., 2022b; Yang et al., 2023), and image (Batzolis et al., 2021; Ho et al., 2022a) generation, and have demonstrated potential across a wide range of domains like computer visions (Baranchuk et al., 2022; Whang et al., 2022), medical image reconstruction (Chung & Ye, 2022; Cao et al., 2024), and bioinformatics (Trippe et al., 2023; Guo et al., 2024). The main goal of diffusion models is to generate novel samples from an unknown and unstructured data distribution. In general, the framework of score-based diffusion models consists of two stochastic processes: a predefined forward noising process that gradually corrupts data into some easy-to-sample noise distribution (e.g., standard Gaussian), and a reverse generative process generating new samples that almost obeys the real data distribution from the pure noise by learning the logarithmic gradient of the forward marginal distributions known as the (Stein) score.

Previous works on diffusion models primarily focus on continuous state spaces, particularly Euclidean space $\mathbb{R}^d$. In this setting, as studied by Song et al. (2021b), the forward process can be constructed as a continuous-time process characterized by a stochastic differential equation (SDE). The associated reverse process is also described by an SDE, which incorporates the drift and diffusion coefficients of the forward SDE along with the score function. Hence, an approximate reverse process can be constructed by estimating the score functions, which is the core principle of the score-based diffusion models. However, many problems require us to deal with discrete data, which arise in fields like text generation (Austin et al., 2021; Hoogeboom et al., 2021; Li et al., 2022;

Zheng et al., 2023), protein design (Gruver et al., 2024; Campbell et al., 2024), and segmentation maps (Zbinden et al., 2023; Courdier et al., 2024). Therefore, modeling probability distributions and constructing diffusion processes in discrete space is of great importance.

To tackle the discrete data, analogous to the state-of-the-art Denoising Diffusion Probabilistic Models (DDPMs) (Ho et al., 2020) in continuous state space, Austin et al. (2021) proposed structured discrete diffusion models in discrete time, which was referred to as Discrete Denoising Diffusion Probabilistic Models (D3PMs). Campbell et al. (2022) introduced a continuous-time framework for discrete diffusion models, in which the forward noising process is formulated as a CTMC. Similar to the continuous SDE setting, the reversal of a forward CTMC can be characterized by the forward rate matrix and the forward marginal probability ratios known as the *discrete score function*. Lou et al. (2024) proposed methods for estimating the discrete score by minimizing discrete score matching objectives. Thus, we can refer to the discrete diffusion models by estimating the score function within the CTMC framework as the *score-based discrete diffusion models*. Interestingly, Benton et al. (2024b) demonstrated that score-based continuous space SDE diffusion and discrete space CTMC diffusion can be regarded as a more general unity, which well illustrates the many similarities between discrete and continuous diffusion.

In light of the significant empirical advances, extensive theoretical works for understanding the efficiency and acceleration of score-based diffusion models in the SDE framework (Chen et al., 2023b; Lee et al., 2022; Yang & Wibisono, 2022; Chen et al., 2023a; Benton et al., 2024a; Li et al., 2023; 2024a; Cheng et al., 2024; Li et al., 2024b; Chen et al., 2024a;b; Gupta et al., 2024) have arisen. Nevertheless, the theoretical foundation remains largely absent for discrete diffusion models, both in discrete-time and continuous-time settings. The recent work Chen & Ying (2024) first studied the theory of score-based discrete diffusion models. This paper proposed a sampling algorithm in the hypercube setting $\{0, 1\}^d$ utilizing the uniformization technique of CTMC, proved its efficiency by deriving convergence bounds and sampling complexities, and applied the elementary properties of the discrete score function to promote the algorithm design. Inspired by these prior works, we study the theoretical analysis of score-based discrete diffusion models within the CTMC framework, as this framework allows us to leverage the score error and its properties in a manner similar to the convergence analysis of continuous diffusion models. The discrete-time sampling algorithm and analysis in our paper are closely related to the previous works on the theory of continuous diffusion models (Chen et al., 2023b; Benton et al., 2024a;b) and discrete diffusion models (Campbell et al., 2022; Chen & Ying, 2024). Our contributions are summarized as follows.

1. Analogous to the exponential integrator (Zhang & Chen, 2022; De Bortoli, 2022; Yang & Wibisono, 2022; Chen et al., 2023a) within the SDE framework, which discretizes the true reverse SDE, we propose a discrete-time sampling algorithm for high-dimensional discrete diffusion tasks with theoretical guarantee within the CTMC framework. Under reasonable assumptions, we derive convergence bounds for scenarios with and without early stopping. When the data distribution is balanced, the bound without early stopping is tighter than the bound with early stopping. Our main KL divergence bounds for convergence consist of three components: score estimation error, discretization error, and truncation error due to insufficient mixing of the forward process, which are analogous to the typical convergence bounds found in continuous diffusion models (Chen et al., 2023a;b; Yang & Wibisono, 2022; Benton et al., 2024a).

2. Technically, we use a Girsanov-based method to analyze the proposed discrete-time sampling algorithm. We study the theoretical properties of discrete score functions such as the movement bound to facilitate our analysis. The discretization error terms in our main bounds are novel in the literature on score-based discrete diffusion models. Also, we derive the exponential convergence of the forward process with uniform mixing in the general state space $[S]^d$. Our KL divergence bounds are nearly linear in the dimension $d$, matching the best result for continuous diffusion models (Benton et al., 2024a) and discrete diffusion models (Chen & Ying, 2024).

**Notation.** We use lowercase letters to denote scalars and boldface lowercase letters to represent vectors. All vectors are considered as column vectors. The $i$-th entry of a vector $\mathbf{x}$ is denoted by $\mathbf{x}^i$, or simply $x^i$ when the context is clear. We use $\mathbf{x}^{\setminus i}$ to refer to all dimensions of $\mathbf{x}$ except the $i$-th, and $\mathbf{x}^{\setminus i} \odot \hat{x}^i$ to denote a vector whose $i$-th dimension takes the value $\hat{x}^i$, while the other dimensions remain as $\mathbf{x}^{\setminus i}$. For a positive integer $n$, we denote $[n]$ as the set $\{1, 2, \ldots, n\}$, $\mathbf{1}_n \in \mathbb{R}^n$ as the vector of ones, and $I_n \in \mathbb{R}^{n \times n}$ as the identity matrix. The notation $\mathbf{e}_i$ refers to a one-hot vector with a 1 in the $i$-th position. $\delta\{\cdot, \cdot\}$ denotes the Kronecker delta, and $\mathbf{1}\{\cdot\}$ is the indicator function. The

Table 1: Comparison of convergence results for score-based discrete diffusion models with and without early stopping. The true reverse CTMC is discretized in our algorithm, leading to a step size $h$ in our bound. For early stopping, a terminal time of $T - \delta$ is used, where $q_\delta$ is a $\delta$-uniform perturbation of the data distribution, and $p_{T-\delta}$ is the law of output from the diffusion model after $K$ steps with $T - \delta = Kh$. Without early stopping, $q_0 = p_{\text{data}}$ is the data distribution, and $p_T$ is the law of output from the diffusion model with time horizon $T$ after $K$ iterations with $T = Kh$. $\epsilon$ is score error, $S$ is the size of discrete state space, $C_1$ depends on $S$ and $\delta$, and $C_2$ and $\kappa^2$ depend on the properties of data distribution. To keep comparison consistent, we present all bounds with $\epsilon$, though in Chen & Ying (2024) the actual bound is $\epsilon T$ due to their time-averaged score error assumption.

| Space | Time-discretized? | Early stopping? | Assumption | Convergence result | Reference |
|---|---|---|---|---|---|
| $\{0,1\}^d$ | No | Yes | continuous score error bounded score estimator | $D_{\text{KL}}(q_\delta \| p_{T-\delta}) \lesssim de^{-T} + \epsilon$ | (Chen & Ying, 2024, Theorem 6) |
| $[S]^d$ | Yes | Yes | discretized score error | $D_{\text{KL}}(q_\delta \| p_{T-\delta}) \lesssim de^{-T} \log S + \delta^{-3} C_1 S^2 h^3 d + C_1 S^2 h^2 dT + C_1^2 \epsilon$ | Theorem 1 (This work) |
| $\{0,1\}^d$ | No | No | continuous score error bounded score estimator bounded score of $p_{\text{data}}$ | $D_{\text{KL}}(q_0 \| p_T) \lesssim de^{-T} + \epsilon$ | (Chen & Ying, 2024, Theorem 7) |
| $[S]^d$ | Yes | No | discretized score error bounded score of $p_{\text{data}}$ | $D_{\text{KL}}(q_0 \| p_T) \lesssim de^{-T} \log S + C_2 S^2 h^2 \kappa^2 T + C_2^2 \epsilon$ | Theorem 2 (This work) |

Hamming distance between two vectors $\mathbf{x}$ and $\mathbf{y}$ is denoted by $\text{Ham}(\mathbf{x}, \mathbf{y}) = \sum_i \mathbf{1}\{x^i \neq y^i\}$. We adopt $f \lesssim g$ to mean that there is a universal constant $C > 0$ such that $f \leq Cg$. Additionally, we denote the generalized I-divergence (Amari, 2012) as $D_I(\mathbf{x} \| \mathbf{y}) = \sum_i (-x^i + y^i + x^i \log(x^i/y^i))$, which is the Bregman divergence with respect to the negative entropy function $I(\mathbf{x}) = \sum_i x^i \log x^i$.

## 2 RELATED WORK

**Discrete Diffusion Models.** The diffusion model was first introduced by Sohl-Dickstein et al. (2015). Many previous works construct discrete diffusion processes as discrete-time Markov chains, and thus train and sample the model in discrete time (Austin et al., 2021; Chen et al., 2024c). In particular, D3PM (Austin et al., 2021) and the Discrete Non-Markov Diffusion Model (DNDM) (Chen et al., 2024c) serve as the discrete diffusion counterparts to DDPM (Ho et al., 2020) and Denoising Diffusion Implicit Model (DDIM) (Song et al., 2021a) in continuous diffusion, respectively. These works offer various practical sampling algorithms but lack theoretical guarantees. Given the limitations and inflexibility of the discrete-time formulation, Campbell et al. (2022) proposed a CTMC framework for discrete diffusion models, which offers much greater flexibility in defining the reverse sampling scheme. Due to the particular nature of the discrete space, this paper sensibly proposed to assume the factorization of the forward process for high-dimensional tasks, a strategy that has been widely adopted, and introduced a tau-leaping method to simulate the continuous-time reverse process. Lou et al. (2024) provided a practical sampling algorithm using Euler discretization and the discrete Tweedie's theorem.

Inspiringly, the CTMC framework for discrete diffusion models makes estimating the discrete score function important for simulating the reverse process. Meng et al. (2022) proposed a concrete score matching objective, but the $L^2$ distance there cannot fully capture the characteristics of the score and is thus unsatisfactory. Sun et al. (2023) derived the score matching objective from the marginal probabilities of each dimension with maximum likelihood training. Lou et al. (2024) proposed score entropy losses by deriving the KL divergence path measure between the true and approximate reverse processes, which are analogous to the score matching objectives in continuous diffusion models (Hyvärinen & Dayan, 2005; Vincent, 2011). Benton et al. (2024b) provided a general framework for these score matching objectives in both discrete and continuous spaces.

**Convergence Analysis of Discrete Diffusion Models.** There is a lack of convergence analysis of discrete diffusion models in the existing literature. Campbell et al. (2022) derived an error bound

for the tau-leaping sampling algorithm with TV distance metric, under strong assumptions such as bounded forward probability ratios and $L^\infty$ error for the approximate rate matrix, since the forward process can be quite general. The error bound grows at least quadratically in the dimension $d$. Chen & Ying (2024) first introduced a sampling algorithm for score-based discrete diffusion models that exactly simulates the reverse process through the uniformization of CTMC in the hypercube setting with independent flips as the forward process. They also provided corresponding convergence bounds and algorithm complexities that are nearly linear in the dimension $d$, matching the best result achieved for continuous diffusion models (Benton et al., 2024a). The results of Chen & Ying (2024) and ours are summarized in Table 1.

## 3 BACKGROUNDS ON SCORE-BASED DISCRETE DIFFUSION MODEL

We will be handling discrete data $x_0 \in \mathcal{X} = [N]$. A probability distribution on $\mathcal{X}$ can be represented by a probability mass vector $p \in \mathbb{R}^N$, where the entries of $p$ are non-negative and sum to 1. Assume that $x_0 \sim p_{\text{data}}$ for some discrete data distribution $p_{\text{data}}$. The forward noising process is defined as a CTMC on $\mathcal{X}$, evolving from $t = 0$ to $t = T$, with a rate matrix (or generator matrix) $Q_t \in \mathbb{R}^{N \times N}$ and an initial distribution $q_0$. Rate matrix $Q_t$ defines the infinitesimal transition probability for the continuous-time process between the two time points $t$ and $t + \Delta t$:

$$q_{t+\Delta t|t}(y|x) = \delta\{x, y\} + Q_t(x, y)\Delta t + o(\Delta t), \tag{1}$$

where $Q_t(x, y)$ is the $(x, y)$ element of the rate matrix $Q_t$ and $q_{t+\Delta t|t}(y|x)$ denotes the infinitesimal transition probability of being in state $y$ at time $t + \Delta t$ given state $x$ at time $t$. From (1) we know

$$Q_t(x, y) \geq 0 \quad \text{for } x \neq y, \quad Q_t(x, x) \leq 0, \quad \text{and} \quad Q_t(x, x) = -\sum_{y \neq x} Q_t(x, y).$$

Moreover, the forward marginal distribution $q_t$ satisfies Kolmogorov forward equation (see, e.g., Campbell et al. (2022); Chewi (2023)):

$$\frac{\mathrm{d}q_t}{\mathrm{d}t} = Q_t^\top q_t, \quad q_0 = p_{\text{data}}. \tag{2}$$

Generally, $Q_t$ is chosen as a simple matrix such that the forward process mixes quickly towards noise distribution $p_{\text{ref}}$ which is easy to sample. For example, if setting $\beta(t)$ as a time-dependent scalar, the uniform rate matrix $Q_t = \beta(t)(\mathbf{1}_N \mathbf{1}_N^\top - N \cdot I_N)$ results in $p_{\text{ref}} = \frac{1}{N}\mathbf{1}_N$, the uniform distribution on $\mathcal{X}$; $Q_t = \beta(t)(\mathbf{1}_N \cdot \mathbf{e}_{\text{MASK}}^\top - I_N)$ yields $p_{\text{ref}} = \mathbf{e}_{\text{MASK}}$, the one-hot probability encoding of the MASK absorbing state (Austin et al., 2021; Campbell et al., 2022; Lou et al., 2024).

Notably, the forward process has an exact time reversal (Kelly, 2011; Campbell et al., 2022) which is also a CTMC that evolves from $t = 0$ to $t = T$ with rate matrix $Q_t^\leftarrow$ given by

$$Q_t^\leftarrow(x, y) = Q_{T-t}(y, x)\frac{q_{T-t}(y)}{q_{T-t}(x)} \quad \text{for } x \neq y, \quad \text{and} \quad Q_t^\leftarrow(x, x) = -\sum_{y \neq x} Q_t^\leftarrow(x, y).$$

Therefore, we know that the access to probability ratio $\frac{q_t(y)}{q_t(x)}$ is important to simulate the reversal. Specifically, the collective ratios $s_t(x) := \left(\frac{q_t(y)}{q_t(x)}\right)_{y \neq x} \in \mathbb{R}^{N-1}$ are known as the discrete score function (Meng et al., 2022), which generalizes the Stein score function $\nabla_x \log q_t(x)$ (Song & Ermon, 2019) in the continuous setting. As pointed out by previous works concerning score matching in discrete space (Lou et al., 2024; Benton et al., 2024b), we learn a discrete score estimator $\hat{s}_t$ to $s_t$ for $t \in [0, T]$ by minimizing the score entropy

$$\mathcal{L}_{\text{SE}}(\hat{s}) = \int_0^T \mathbb{E}_{x_t \sim q_t} \sum_{y \neq x_t} Q_t(y, x_t) D_I(s_t(x_t)_y \| \hat{s}_t(x_t)_y) \, \mathrm{d}t.$$

Note that the score entropy loss is characterized by the Bregman divergence, different from the $L^2$ distance loss in the continuous counterpart. We defer the details on the Bregman divergence to Appendix E. The score entropy is exactly the path measure KL divergence (Lou et al., 2024; Chen & Ying, 2024; Benton et al., 2024b). Although the score entropy can not be directly estimated, there are equivalent objectives such as implicit score entropy and denoising score entropy (Lou et al., 2024; Benton et al., 2024b) which can be optimized practically.

## 4 PROBLEM SETTING

In practice, the discrete state space typically factorizes as $\mathcal{X} = [S]^d$, representing sequences $\mathbf{x} = \mathbf{x}^{1:d} = x^1 \cdots x^d$ (e.g., text token sequences (Lou et al., 2024), image pixel values (Campbell et al., 2022), or protein sequences (Campbell et al., 2024)).

**Forward Process.** The forward process $X = (X_t)_{t \geq 0}$ is defined by a CTMC on $\mathcal{X}$ with rate matrix $Q_t$ starting from $X_0 \sim q_0 = p_{\text{data}}$ which is the data distribution. Let $q_t := \text{Law}(X_t)$, which follows the Kolmogorov forward equation (2). As a general $Q_t \in \mathbb{R}^{S^d \times S^d}$ would be of exponential size, we assume that the forward process can be factorized such that each dimension propagates independently with rate $Q_t^{\text{tok}} \in \mathbb{R}^{S \times S}$. Namely, the forward transition kernel $q_{t|s}(\mathbf{x}_t|\mathbf{x}_s) = \mathbb{P}(X_t = \mathbf{x}_t|X_s = \mathbf{x}_s)$ factorizes as $q_{t|s}(\mathbf{x}_t|\mathbf{x}_s) = \prod_{i=1}^d q_{t|s}^i(x_t^i|x_s^i)$, where $q_{t|s}^i(x_t^i|x_s^i) = \mathbb{P}(X_t^i = x_t^i|X_s^i = x_s^i)$ is the transition probability for the $i$-th dimensional CTMC $X^i := (X_t^i)_{t \geq 0}$ with forward rate $Q_t^{\text{tok}}$. This is a common assumption in literature considering high dimension tasks in the CTMC setting (Campbell et al., 2022; Lou et al., 2024; Campbell et al., 2024; Chen & Ying, 2024). According to Campbell et al. (2022, Proposition 3), the non-zero off-diagonal entries of $Q_t$ are given by

$$Q_t(\mathbf{x}, \mathbf{x}^{\backslash i} \odot \hat{x}^i) = Q_t(x^1 \cdots x^i \cdots x^d, x^1 \cdots \hat{x}^i \cdots x^d) = Q_t^{\text{tok}}(x^i, \hat{x}^i)$$

for $\mathbf{x} \in \mathcal{X}$ and $x^i \neq \hat{x}^i \in [S]$. We see that the non-zero off-diagonal entries of $Q_t$ can only occur between sequences with a Hamming distance of 1, leading to a rather sparse structure. As for the perturbation of each dimension, we take the time-homogeneous uniform rate

$$Q_t^{\text{tok}} \equiv Q^{\text{tok}} = \frac{1}{S}\mathbf{1}_S\mathbf{1}_S^\top - I_S.$$

This rate matrix is common in applications (Austin et al., 2021; Campbell et al., 2022; Lou et al., 2024; Campbell et al., 2024), and also analyzed by Chen & Ying (2024) where $S = 2$ is taken. Then $Q_t \equiv Q$ is of the form

$$Q(\mathbf{x}, \mathbf{y}) = \begin{cases} \frac{1}{S}, & \text{Ham}(\mathbf{x}, \mathbf{y}) = 1, \\ (\frac{1}{S} - 1)d, & \mathbf{x} = \mathbf{y}, \\ 0, & \text{otherwise}. \end{cases}$$

We can then obtain the expression of forward transition probabilities and marginals as stated in the following proposition, showing that the marginal converges to a uniform distribution over $\mathcal{X} = [S]^d$ which we denote $\pi^d$ as $t$ increases. The proof of Proposition 1 is deferred to Appendix A.1.

**Proposition 1.** *Let $P_{s,t}^i \in \mathbb{R}^{S \times S}$ be the transition probability matrix of the $i$-th dimensional forward CTMC $X^i$ from time $s$ to time $t$, i.e., $P_{s,t}^i(x, y) = q_{t|s}^i(y|x)$ for all $x, y \in [S]$ and $i \in [d]$. Then for all $i \in [d]$, $P_{s,t}^i \equiv P_{s,t}^0$ where*

$$P_{s,t}^0 = \frac{1}{S}(1 - e^{-(t-s)})\mathbf{1}_S\mathbf{1}_S^\top + e^{-(t-s)}I_S.$$

*Let $P_{s,t} \in \mathbb{R}^{S^d \times S^d}$ be the transition probability matrix of the forward process from time $s$ to time $t$ for $t > s$, i.e., $P_{s,t}(\mathbf{x}, \mathbf{y}) = q_{t|s}(\mathbf{y}|\mathbf{x})$ for all $\mathbf{x}, \mathbf{y} \in \mathcal{X}$, then*

$$P_{s,t} = \left(P_{s,t}^0\right)^{\otimes d},$$

*where $(\cdot)^{\otimes d}$ denotes performing the matrix Kronecker product $d$ times. In particular, the marginal distribution of the forward process at time $t$ can be expressed as*

$$q_t = \left[\frac{1}{S}(1 - e^{-t})\mathbf{1}_S\mathbf{1}_S^\top + e^{-t}I_S\right]^{\otimes d} \cdot p_{\text{data}},$$

*and when $t \to +\infty$, $q_t$ approaches the uniform distribution $\pi^d$.*

**Reverse Process and Score Model.** Suppose that we run the forward process until time $T > 0$, ending at $q_T$. We can convert the noise back into samples if we reverse the forward CTMC dynamic $X$ in time. According to Campbell et al. (2022, Proposition 3), the reverse process $Y = (Y_t)_{t \in [0,T]}$

can be achieved by a CTMC starting from $Y_0 \sim q_T$ with $\mathrm{Law}(Y_t) \overset{\mathrm{a.s.}}{=} \mathrm{Law}(X_{T-t}) = q_{T-t}$, and the reverse rate $Q_t^{\leftarrow} \in \mathbb{R}^{S^d \times S^d}$ is of the form

$$Q_t^{\leftarrow}(\mathbf{x}, \tilde{\mathbf{x}}) = \sum_{i=1}^{d} Q_{T-t}^{\mathrm{tok}}(\tilde{\mathbf{x}}^i, \mathbf{x}^i)\delta\{\mathbf{x}^{\backslash i}, \tilde{\mathbf{x}}^{\backslash i}\}\frac{q_{T-t}(\tilde{\mathbf{x}})}{q_{T-t}(\mathbf{x})}. \tag{3}$$

From (3) we know that the non-zero off-diagonal entries of $Q_t^{\leftarrow}$ can only occur between sequences with a Hamming distance of 1. Therefore, we care about the forward marginal ratios between these sequences and denote the collective ratios as $s_t : \mathcal{X} \to \mathbb{R}^{d(S-1)}$ defined by

$$s_t(\mathbf{x})_{i,\hat{x}^i} := \frac{q_t(\mathbf{x}^{\backslash i} \odot \hat{x}^i)}{q_t(\mathbf{x})} \quad \text{for } \mathbf{x} \in \mathcal{X}, \ i \in [d], \ x^i \neq \hat{x}^i \in [S].$$

Then the sparse structure of $Q_t^{\leftarrow}$ can be expressed as

$$Q_t^{\leftarrow}(\mathbf{x}, \mathbf{x}^{\backslash i} \odot \hat{x}^i) = Q_{T-t}^{\mathrm{tok}}(\hat{x}^i, x^i)s_{T-t}(\mathbf{x})_{i,\hat{x}^i} = \frac{1}{S}s_{T-t}(\mathbf{x})_{i,\hat{x}^i} \quad \text{for } i \in [d], \ x^i \neq \hat{x}^i \in [S],$$

and the reverse marginal $q_{T-t}$ satisfies the Kolmogorov equation

$$\frac{\mathrm{d}q_{T-t}}{\mathrm{d}t} = Q_t^{\leftarrow\top}q_{T-t}. \tag{4}$$

However, in practice, we do not have access to $q_T$, the initial distribution of the reverse process, so we start the reverse process at the noise $\pi^d$, the target distribution of the forward process, and simulate it with score estimators. The score estimator $\hat{s}_t : \mathcal{X} \to \mathbb{R}^{d(S-1)}$ that estimates $s_t$ for $t \in [0, T]$ is learned by minimizing the score entropy loss

$$\mathcal{L}_{\mathrm{SE}}(\hat{s}) = \int_0^T \mathbb{E}_{\mathbf{x}_t \sim q_t} \sum_{i=1}^{d} \sum_{\hat{x}_t^i \neq x_t^i} Q_t(\mathbf{x}_t^{\backslash i} \odot \hat{x}_t^i, \mathbf{x}_t)D_I(s_t(\mathbf{x}_t)_{i,\hat{x}_t^i} \| \hat{s}_t(\mathbf{x}_t)_{i,\hat{x}_t^i}) \, \mathrm{d}t$$

$$= \frac{1}{S}\int_0^T \mathbb{E}_{\mathbf{x}_t \sim q_t} D_I(s_t(\mathbf{x}_t) \| \hat{s}_t(\mathbf{x}_t)) \, \mathrm{d}t. \tag{5}$$

**Algorithm.** Since the reverse process is time-inhomogeneous, we can discretize the time to simulate it. Let $h > 0$ be the step size, and $T$ be the time horizon. We consider applying early stopping with a terminal time of $T - \delta$, because the score function can blow up as $t \to 0$ for data distributions without full support on $\mathcal{X}$. As shown in Section 5, early stopping can be removed when $p_{\mathrm{data}}$ has full support on $\mathcal{X}$. The time horizon is set to $T = Kh + \delta$, where $\delta \geq 0$ is a small value and $K \in \mathbb{N}$ is assumed. Suppose we have access to the score estimators $\hat{s}_{T-kh}$ for $k = 0, 1 \cdots, K - 1$. We then construct a continuous-time sampling process $Z = (Z_t)_{t \in [0, T-\delta]}$ starting from $Z_0 \sim \pi^d$, and let $p_t := \mathrm{Law}(Z_t)$. For $t \in [kh, (k+1)h]$, by freezing the value of the rate matrix in the ODEs (4) at time $kh$ and replacing the true score with the score estimator, $(Z_t)_{t \in [kh,(k+1)h]}$ is constructed as a time-homogeneous CTMC with rate matrix $\hat{Q}_{kh}^{\leftarrow} \in \mathbb{R}^{S^d \times S^d}$, where the non-zero off-diagonal entries of $\hat{Q}_{kh}^{\leftarrow}$ are given by

$$\hat{Q}_{kh}^{\leftarrow}(\mathbf{x}, \mathbf{x}^{\backslash i} \odot \hat{x}^i) = \frac{1}{S}\hat{s}_{T-kh}(\mathbf{x})_{i,\hat{x}^i} \quad \text{for } i \in [d], \ x^i \neq \hat{x}^i \in [S].$$

Hence, in each step from time $kh$ to $(k+1)h$, the distribution follows the Kolmogorov equation

$$\frac{\mathrm{d}}{\mathrm{d}t}p_{kh+t|kh}(\mathbf{x}_{kh+t}|\mathbf{x}_{kh}) = \hat{Q}_{kh}^{\leftarrow\top}p_{kh+t|kh}(\mathbf{x}_{kh+t}|\mathbf{x}_{kh}), \ t \in [0, h]. \tag{6}$$

For $k = 0, 1 \cdots, K - 1$, theoretically we update the solution of the ODEs (6) as

$$\mathbf{z}_{k+1} \sim \exp\left(h\hat{Q}_{hk}^{\leftarrow\top}\right) \cdot (\mathbf{e}_{\mathbf{z}_k^1} \otimes \cdots \otimes \mathbf{e}_{\mathbf{z}_k^d}), \ \mathbf{z}_0 \sim \pi^d. \tag{7}$$

After $K$ iterations, we get a sample $\mathbf{z}_K$ with law $p_{T-\delta}$. Unlike the exponential integrator in the continuous setting, where the closed-form solution for the iteration formula of sampling random variables can be obtained by solving the linear discretized SDE within each time interval, the discrete setting requires deriving the categorical probability from the Kolmogorov equation and then sampling from this distribution. In practice for sampling, we can run (7) with a Poisson point process utilizing the uniformization of CTMC (Chen & Ying, 2024) instead of exactly calculating the matrix exponential $\exp(h\hat{Q}_{hk}^{\leftarrow\top})$ which is intractable for reasonably sized $S$ and $d$. For completeness, we formalize the practical sampling procedure in Algorithm 1, presented in Appendix D.

## 5   MAIN RESULTS

Our main results are the convergence analyses for the theoretical iterative algorithm (7). We use an analogous Girsanov-based method in the continuous SDE settings (Chen et al., 2023a;b; Benton et al., 2024a) by bounding the KL divergence between the path measures of the true reverse process and the sampling process.

The score estimation error measures the quality of the learned score estimator. Since we discretize the true reverse CTMC in our algorithm, we make the following score error assumption.

**Assumption 1** (Score estimation error). *The score estimator satisfies*

$$\frac{1}{S}\sum_{k=0}^{K-1}\int_{kh+\delta}^{(k+1)h+\delta}\mathbb{E}_{\mathbf{x}_t\sim q_t}D_I\big(s_{(k+1)h+\delta}(\mathbf{x}_t)\|\hat{s}_{(k+1)h+\delta}(\mathbf{x}_t)\big)\,\mathrm{d}t\leq\epsilon_{\mathrm{score}}.$$

Assumption 1, which we introduce for the first time, can be viewed as a time-discretization of the score entropy loss (5), establishing a discretized version of score error assumptions for our CTMC framework. This assumption is analogous to those widely used in diffusion models (Lee et al., 2022; Chen et al., 2023a;b; Li et al., 2023; 2024a;b; Benton et al., 2024a; Chen & Ying, 2024), but with two key distinctions. First, we use the Bregman distance instead of the $L^2$ distance, aligning with the discrete nature of the CTMC framework. Second, in contrast to Chen & Ying (2024), we only require a small error over pre-defined discretization points, rather than a continuous error bound.

**Theorem 1.** *Suppose Assumption 1 holds. By choosing a small $\delta = \tilde{O}(S^{-\frac{2}{3}}) > 0$, the KL divergence between $q_\delta$ and $p_{T-\delta}$ is bounded by*

$$D_{\mathrm{KL}}(q_\delta\|p_{T-\delta}) \lesssim de^{-T}\log S + \delta^{-3}C_1 S^2 h^3 d + C_1 S^2 h^2 dT + C_1^2\epsilon_{\mathrm{score}}, \qquad (8)$$

*where $C_1 = \max\left\{1+\frac{S}{\delta}, \max_{\mathbf{x}\in\mathcal{X},k\in\{0,\cdots,K-1\}}\|\hat{s}_{(k+1)h+\delta}(\mathbf{x})\|_\infty\right\}$, and the TV distance between $p_{\mathrm{data}}$ and $p_{T-\delta}$ is bounded by*

$$D_{\mathrm{TV}}(p_{\mathrm{data}},p_{T-\delta}) \lesssim \sqrt{de^{-T}\log S + \delta^{-3}C_1 S^2 h^3 d + C_1 S^2 h^2 dT + C_1^2\epsilon_{\mathrm{score}}}+(1-e^{-d\delta(S-1)/S}). \tag{9}$$

The proof of Theorem 1 is deferred to Appendix A.3. We interpret the KL divergence bound (8) as follows. The first term $de^{-T}\log S$ arises from the initialization error of the algorithm. Recall that the true reverse process should begin from $q_T$, but the algorithm starts from the noise $\pi^d$. The second term $\delta^{-3}C_1 S^2 h^3 d + C_1 S^2 h^2 dT$ reflects the discretization error, which scales with the step size $h$ and is linear in $d$, vanishing as $h \to 0$. The third term $C_1^2\epsilon_{\mathrm{score}}$ corresponds to the score estimation error, which is non-vanishing. We remark that the appearance of the quantity $C_1$ in the bound (8), which is absent in the continuous SDE counterpart, stems from the lack of the triangle inequality for the Bregman divergence. Specifically, $C_1$ is the uniform bound of the involved score and score estimators (see Lemma 2 in Appendix for details). As discussed in Appendix A.4, if applying the score clipping technique, we can ensure that $C_1 \lesssim S/\delta$. The nearly linear dependence on $d$ of the KL bound (8) matches the best result for the continuous diffusion model (Benton et al., 2024a).

The last term in (9) provides an upper bound on the TV distance between $p_{\mathrm{data}}$ and $q_\delta$. There is a trade-off involving $\delta$ in this bound: to reduce $D_{\mathrm{TV}}(p_{\mathrm{data}}, q_\delta)$, we opt for a rather small $\delta > 0$, especially when $d$ is large. However, this causes the square root term to grow rapidly, at a rate of $\delta^{-2}$. As a result, the square root term dominates the bound, leading us to focus on the KL divergence bound (8) for a small $\delta$.

The data distribution characteristics are closely related to the properties of the score function when $t > 0$ is small. We perform early stopping because the score of the data distribution can be positive infinity for some data points with a probability of zero. However, if the score of $p_{\mathrm{data}}$ is uniformly bounded, early stopping is no longer necessary.

**Assumption 2.** *The data distribution $p_{\mathrm{data}}$ has full support on $\mathcal{X}$, and there exists a uniform $L > 0$ depending on $S$ but not on $d$, such that for all $\mathbf{x} \in \mathcal{X}, i \in [d]$, and $x^i \neq \hat{x}^i \in [S]$,*

$$s_0(\mathbf{x})_{i,\hat{x}^i} = \frac{p_{\mathrm{data}}(\mathbf{x}^{\backslash i}\odot\hat{x}^i)}{p_{\mathrm{data}}(\mathbf{x})} \leq L.$$

Assumption 2 holds in many cases, e.g., when the data distribution is the product of i.i.d. components, each with a marginal distribution fully supported on $[S]$. It is similar to Assumption 3 in Chen & Ying (2024), both aimed to remove early stopping. Our Assumption 2 is stronger since we assume that $L$ does not depend on $d$. Campbell et al. (2022) also applied similar assumptions, as presented in Assumptions 1 and 2 in their paper. As they noted, the uniform boundness for the data distribution follows trivially from the strict positiveness of $p_{\text{data}}$ if we allow $L$ to depend on $d$. We adopt the assumption from Campbell et al. (2022), which enables us to derive a bound that can be nearly linear in $d$, as stated in the following theorem.

**Theorem 2.** *Suppose Assumptions 1 and 2 hold. Let $\delta = 0$. Denote $p_{\text{data}}^i \in \mathbb{R}^S$ as the marginal distribution of the $i$-th dimension of the data, and let $\kappa_i = \frac{(p_{\text{data}}^i)_{\max}}{(p_{\text{data}}^i)_{\min}}$ for $i \in [d]$ and $\kappa^2 = \sum_{i=1}^d \kappa_i^2$, then it holds that*

$$D_{\text{KL}}(p_{\text{data}} \| p_T) \lesssim de^{-T} \log S + C_2 \kappa^2 S^2 h^2 T + C_2^2 \epsilon_{\text{score}}, \tag{10}$$

*where $C_2 = \max \left\{ L, \max_{\mathbf{x} \in \mathcal{X}, k \in \{0, \cdots, K-1\}} \|\hat{s}_{(k+1)h}(\mathbf{x})\|_\infty \right\}$.*

The proof of Theorem 2 is deferred to Appendix A.4. $\kappa_i$ is well-defined since the strict positiveness of $p_{\text{data}}$ in Assumption 2 ensures the strict positiveness of marginal distribution $p_{\text{data}}^i$. In the bound (10), the term $C_2 \kappa^2 S^2 h^2 T$ is the discretization error, where $\kappa^2$ and $C_2$ are related to the property of the data distribution. Specifically, $\kappa^2$ is characterized by the ratio of the largest to the smallest entry in the marginal data distribution, and a large value of $\kappa^2$ indicates that the probability values for certain data points are either very high or very low. $\kappa^2$ is nearly linear in $d$ and equals $d$ when $p_{\text{data}}$ is a uniform distribution. When $p_{\text{data}}$ is relatively balanced, meaning that the probabilities across data points are fairly similar, $\kappa^2/d$ and $C_2$ are reasonably small, and thus the bound (10) is tighter than (8) as it eliminates the term involving $\delta^{-4}$. The quantity $C_2$ arises similarly to $C_1$, and applying score clipping can make $C_2 \lesssim L$ as shown in Appendix A.4, where $L$ is proven to be the uniform bound of the score along the forward process.

With the early stopping criterion, Theorem 1 results in the following iteration complexity.

**Corollary 1.** *Suppose Assumption 1 holds. By choosing a small $\delta = \tilde{O}(S^{-2/3}) > 0$, for any $\epsilon > 0$, if choosing $T \asymp \log \left( \frac{d \log S}{\epsilon} \right)$ and $h \asymp \min \left\{ \delta \left( \frac{\epsilon}{C_1 S^2 d} \right)^{1/3}, \left( \frac{\epsilon}{C_1 S^2 d} \right)^{1/2} \right\}$, then the discrete diffusion model requires at most $\tilde{O} \left( \max \left\{ \frac{1}{\delta} \left( \frac{C_1 S^2 d}{\epsilon} \right)^{1/3}, \left( \frac{C_1 S^2 d}{\epsilon} \right)^{1/2} \right\} \right)$ steps to reach a distribution $p_{T-\delta}$ with $D_{\text{KL}}(q_\delta \| p_{T-\delta}) = \tilde{O}(\epsilon + C_1^2 \epsilon_{\text{score}})$.*

By Corollary 1, to have $D_{\text{KL}}(q_\delta \| p_{T-\delta}) = \tilde{O}(\epsilon)$, it suffices to choose $\epsilon_{\text{score}} = O(\epsilon/C_1^2)$. Similarly, Theorem 2 leads to the following iteration complexity without early stopping.

**Corollary 2.** *Suppose Assumptions 1 and 2 hold. Let $\delta = 0$. For any $\epsilon > 0$, if choosing $T \asymp \log \left( \frac{d \log S}{\epsilon} \right)$ and $h \asymp \sqrt{\frac{\epsilon}{C_2 S^2 \kappa^2}}$, then the discrete diffusion model requires at most $\tilde{\Theta} \left( \sqrt{\frac{C_2 S^2 \kappa^2}{\epsilon}} \right)$ steps to reach a distribution $p_T$ with $D_{\text{KL}}(p_{\text{data}} \| p_T) \leq \tilde{O}(\epsilon + C_2^2 \epsilon_{\text{score}})$.*

By Corollary 2, to have $D_{\text{KL}}(p_{\text{data}} \| p_T) = \tilde{O}(\epsilon)$, it suffices to have $\epsilon_{\text{score}} = O(\epsilon/C_2^2)$.

**Comparison with Chen & Ying (2024).** Both Chen & Ying (2024) and our paper utilize a uniform rate matrix for its structural simplicity. Chen & Ying (2024) applied the uniformization technique of CTMC to develop a sampling algorithm for discrete diffusion models, enabling the exact simulation of approximate reverse CTMC dynamics. In contrast, our algorithm discretizes the time to simulate the reverse process and calls the score estimator at fixed discretization points $\{kh + \delta\}_{k \in [K]}$ instead of at randomly sampled times, introducing an additional discretization error term in our bounds. As $h \to 0$, our sampling procedure covers the exact simulation, at which point our bound (8) degrades to the bound in terms of score error and the mixing of the forward noising process. See Table 1 for a comparison of the quantitative KL divergence bounds. Moreover, although we do not explicitly make additional assumptions on the score estimator, we note that the score clipping approach discussed in Appendices A.4 and D aligns with Assumption 2 in Chen & Ying (2024), which assumes a bounded score estimator to implement their sampling algorithm. Furthermore, our convergence analysis is conducted in the more general $[S]^d$ setting, as opposed to the $\{0, 1\}^d$ hypercube framework used in their work, making our state space setting and algorithm more broadly applicable.

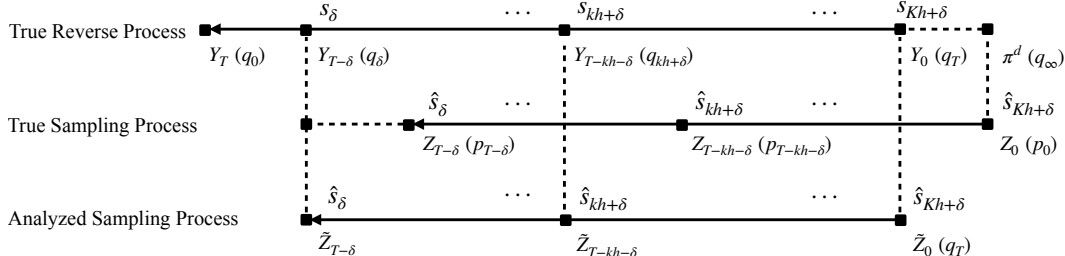

Figure 1: Illustration of the processes $Y$, $Z$ and $\tilde{Z}$. $Y$ is the true reverse process, $Z$ is the sampling process starting from the noise $\pi^d$, and $\tilde{Z}$ is the same as $Z$ except for the initialization. Both $Y$ and $\tilde{Z}$ start from $q_T$. The random variables (e.g., $Y_t$, $Z_t$) are shown with their corresponding probability laws in parentheses (e.g., $q_{T-t}$, $p_t$).

## 6 OVERVIEW OF KEY PROOF TECHNIQUES

We present key proof techniques for the main results given in Section 5, which characterize the discrete-time sampling process in the discrete diffusion model. Theorems 1 and 2 provide bounds for both the KL divergence and the TV distance. In this section, we focus on proving the KL bound (8), as the TV distance bound (9) follows from this KL bound via Pinsker's inequality, plus the additional term $(1 - e^{-d\delta(S-1)/S})$. The proof of the KL bound (10) follows similarly.

Let $\mathbb{Q}$ be the path measure of the reverse process $Y = (Y_t)_{t \in [0,T]}$ starting from $Y_0 \sim q_T$, and $\mathbb{P}^{\pi^d}$ be the path measure of the sampling process $Z = (Z_t)_{t \in [0,T-\delta]}$ starting from $Z_0 \sim \pi^d$. Recall that $q_t = \mathrm{Law}(Y_{T-t})$ and $p_t = \mathrm{Law}(Z_t)$. Additionally, let the process $\tilde{Z} = (\tilde{Z}_t)_{t \in [0,T-\delta]}$ be the same as $Z$ except for its initialization at $q_T$, and $\mathbb{P}^{q_T}$ be the path measure of $\tilde{Z}$. We visualize these three processes in Figure 1. Then by the data processing inequality and the chain rule of KL divergence, we have

$$D_{\mathrm{KL}}(q_\delta \| p_{T-\delta}) \le D_{\mathrm{KL}}(\mathbb{Q} \| \mathbb{P}^{\pi^d}) = D_{\mathrm{KL}}(q_T \| \pi^d) + D_{\mathrm{KL}}(\mathbb{Q} \| \mathbb{P}^{q_T}). \tag{11}$$

Therefore, it suffices to bound the two KL divergence terms in (11).

**Forward Process with General $S$ States.** The term $D_{\mathrm{KL}}(q_T \| \pi^d)$ represents the prior loss due to the initial distribution mismatch between $q_T$ and $\pi^d$. This can be bounded using the convergence properties of the forward process. Proposition 1 already shows that the forward process $q_T$ converges to $\pi^d$ as $T \to \infty$. We now characterize this convergence rate for general $S$ states.

**Proposition 2.** *For the forward process marginal $q_t$ targeting the uniform distribution $\pi^d$, we have*

$$D_{\mathrm{KL}}(q_t \| \pi^d) \le e^{-t} D_{\mathrm{KL}}(p_{\mathrm{data}} \| \pi^d) \le e^{-t} d \log S.$$

The proof of Proposition 2 is deferred to Appendix A.2. Proposition 2 demonstrates that the forward marginal converges exponentially to the uniform distribution $\pi^d$. Since $D_{\mathrm{KL}}(p_{\mathrm{data}} \| \pi^d)$ can be upper bounded by $d \log S$, which is the maximum possible KL divergence between any discrete distribution and the uniform distribution over $S$ states in $d$ dimensions, this result holds irrespective of the complexity of the data distribution $p_{\mathrm{data}}$.

**Girsanov-Based Method.** The term $D_{\mathrm{KL}}(\mathbb{Q} \| \mathbb{P}^{q_T})$ is the discretization error that calculates the path measure KL divergence between the true reverse process and discretized reverse sampling process. We employ Girsanov's theorem to explicitly express the discretization error as follows (detailed in Lemma 1 in Appendix)

$$D_{\mathrm{KL}}(\mathbb{Q} \| \mathbb{P}^{q_T}) = \frac{1}{S} \sum_{k=0}^{K-1} \int_{kh+\delta}^{(k+1)h+\delta} \mathbb{E}_{\mathbf{x}_t \sim q_t} D_I(s_t(\mathbf{x}_t) \| \hat{s}_{(k+1)h+\delta}(\mathbf{x}_t)) \, \mathrm{d}t,$$

where $D_I(s_t(\mathbf{x}_t) \| \hat{s}_{(k+1)h+\delta}(\mathbf{x}_t))$ is the Bregman divergence characterizing the distance between $s_t$ and $\hat{s}_{(k+1)h+\delta}$, for $t \in [kh+\delta, (k+1)h+\delta]$. Since our Assumption 1 is made on the discrete points

$\{kh + \delta\}_{k \in [K]}$, we further decompose the Bregman divergence into two parts:

$$D_{\mathrm{KL}}(\mathbb{Q}\|\mathbb{P}^{q_T}) \lesssim \underbrace{\frac{C_1}{S} \sum_{k=0}^{K-1} \int_{kh+\delta}^{(k+1)h+\delta} \mathbb{E}_{\mathbf{x}_t \sim q_t} \|s_t(\mathbf{x}_t) - s_{(k+1)h+\delta}(\mathbf{x}_t)\|_2^2 \, \mathrm{d}t}_{\text{Score Movement}}$$

$$+ \underbrace{\frac{C_1^2}{S} \sum_{k=0}^{K-1} \int_{kh+\delta}^{(k+1)h+\delta} \mathbb{E}_{\mathbf{x}_t \sim q_t} D_I(s_{(k+1)h+\delta}(\mathbf{x}_t)\|\hat{s}_{(k+1)h+\delta}(\mathbf{x}_t)) \, \mathrm{d}t,}_{\text{Score Error}} \quad (12)$$

where the inequality holds due to the boundness of score and the property of Bregman divergence (detailed in Lemma 2 and Proposition 3 in Appendix). The score movement term in (12) represents the squared norm of the score difference $\|s_t(\mathbf{x}) - s_{(k+1)h+\delta}(\mathbf{x})\|_2^2$ over the time interval $[kh+\delta, (k+1)h + \delta]$, and the score error term in (12) is naturally bounded by $C_1^2 \epsilon_{\mathrm{score}}$ due to Assumption 1.

**Score Movement Bound.** We derive the score movement bound to quantify the change in the score function over time (detailed in Lemma 3 in Appendix). For all $k \in \{0, 1, \cdots, K - 1\}$, $t \in [kh + \delta, (k + 1)h + \delta]$, and $\mathbf{x}_t \in \mathcal{X}$, we establish the following inequality:

$$\frac{1}{S}\|s_t(\mathbf{x}_t) - s_{(k+1)h+\delta}(\mathbf{x}_t)\|_2^2 \lesssim \left[\frac{e^{-(kh+\delta)}}{(1 - e^{-(kh+\delta)})^2} + \frac{S}{e^{kh+\delta} - 1} + 1\right]^2 dS^2 h^2,$$

which leads to the score movement term in (12) being bounded by $\delta^{-3} C_1 dS^2 h^3 + C_1 dS^2 h^2 T$ (detailed in Lemma 2 and the proof of Theorem 1 in Appendix), vanishing as $h$ approaches zero.

In summary, we established proof techniques for bounding (11). We demonstrated exponential convergence of the forward process to the uniform distribution, bounding the prior loss term. The Girsanov-based method was employed to analyze the discretization error, decomposing it into score movement and score error components. The derived score movement bound revealed dependencies on step size $h$, dimension $d$, and state space size $S$. Collectively, these techniques provide a final bound on $D_{\mathrm{KL}}(q_\delta\|p_{T-\delta})$, offering a rigorous framework for analyzing discrete diffusion models under various parameter regimes.

# 7 CONCLUSION AND FUTURE WORK

We introduce a discrete-time sampling algorithm for the high-dimensional score-based discrete diffusion models within the CTMC framework and corresponding convergence analyses using a Girsanov-based method, similar to that in the continuous SDE setting. We study the properties of the discrete score function and incorporate them into our discretization error analysis. The convergence bounds for the sampling algorithm are nearly linear in the dimension $d$, both with and without early stopping. The bound without early stopping is related to the property of the data distribution.

The primary limitation of this work is that the discretization error in the convergence bounds of the proposed sampling algorithm becomes significant when $\delta$ is sufficiently small or when the data distribution contains extreme probability values. Hence, future work can be focused on developing refined techniques to treat the discretization error. Another important future direction will be developing accelerated algorithms with theoretical guarantees for score-based discrete diffusion models and applying our method to some more general rate matrices to obtain tight convergence bounds.

## ACKNOWLEDGEMENTS

We thank the anonymous reviewers and area chair for their helpful comments. ZC and QG are supported in part by the NSF grant IIS-2008981 and Sloan Research Fellowship. ZC is also supported by UCLA Dissertation Year Fellowship. The views and conclusions contained in this paper are those of the authors and should not be interpreted as representing any funding agencies.

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

## A  MAIN PROOFS

### A.1  PROOF OF PROPOSITION 1

*Proof of Proposition 1.* Consider the Kolmogorov forward equation $\frac{\partial}{\partial t}P^i_{s,t} = P^i_{s,t}Q^{\text{tok}}$, we obtain that $P^i_{s,t} = \exp\left((t-s)Q^{\text{tok}}\right)$ for the CTMC $X^i$. Let $Q^{\text{tok}} = P\Lambda P^{-1}$ be the eigendecomposition of $Q^{\text{tok}}$ with

$$
P = \begin{pmatrix} -1 & -1 & \cdots & -1 & 1 \\ 1 & 0 & \cdots & 0 & 1 \\ 0 & 1 & \cdots & 0 & 1 \\ \vdots & \vdots & \ddots & \vdots & \vdots \\ 0 & 0 & \cdots & 1 & 1 \end{pmatrix}, \quad P^{-1} = \frac{1}{S}\begin{pmatrix} -1 & S-1 & \cdots & -1 & -1 \\ \vdots & \vdots & \ddots & \vdots & \vdots \\ -1 & -1 & \cdots & S-1 & -1 \\ -1 & -1 & \cdots & -1 & S-1 \\ 1 & 1 & \cdots & 1 & 1 \end{pmatrix},
$$

and $\Lambda = \mathrm{diag}\{-1, \cdots, -1, 0\}$. Then it can be easily verified that

$$
P^0_{s,t} = \exp\left((t-s)Q^{\text{tok}}\right) = P\exp\left((t-s)\Lambda\right)P^{-1} = \frac{1}{S}\left[(1 - e^{-(t-s)})\mathbf{1}_S\mathbf{1}_S^\top + Se^{-(t-s)}I_S\right].
$$

Also, the Kolmogorov forward equation $\frac{\partial}{\partial t}P_{s,t} = P_{s,t}Q$ yields $P_{s,t} = \exp((t-s)Q)$. Furthermore, note that $Q$ has a Kronecker-product structure

$$
Q = \bigoplus_{k=1}^{d} Q^{\text{tok}} = \sum_{k=1}^{d} I_S^{\otimes(k-1)} \otimes Q^{\text{tok}} \otimes I_S^{\otimes(d-k)}
$$

where $\oplus$ denotes the Kronecker sum, and this structure can be verified by direct calculation:

$$
\begin{aligned}
\left[\bigoplus_{k=1}^{d} Q^{\text{tok}}\right](\mathbf{x}, \mathbf{x}^{\backslash i} \odot \hat{x}^i) &= \sum_{k=1}^{d}\left[I_S^{\otimes(k-1)} \otimes Q^{\text{tok}} \otimes I_S^{\otimes(d-k)}\right](\mathbf{x}, \mathbf{x}^{\backslash i} \odot \hat{x}^i) \\
&= \left[I_S^{\otimes(i-1)} \otimes Q^{\text{tok}} \otimes I_S^{\otimes(d-i)}\right](\mathbf{x}, \mathbf{x}^{\backslash i} \odot \hat{x}^i) \\
&= \prod_{j\neq i} I_S(x^j, x^j) \cdot Q^{\text{tok}}(x^i, \hat{x}^i) = \frac{1}{S} = Q(\mathbf{x}, \mathbf{x}^{\backslash i} \odot \hat{x}^i)
\end{aligned}
$$

and

$$
\begin{aligned}
\left[\bigoplus_{k=1}^{d} Q^{\text{tok}}\right](\mathbf{x}, \mathbf{x}) &= \sum_{k=1}^{d}\left[I_S^{\otimes(k-1)} \otimes Q^{\text{tok}} \otimes I_S^{\otimes(d-k)}\right](\mathbf{x}, \mathbf{x}) \\
&= \sum_{k=1}^{d}\prod_{j\neq k} I_S(x^j, x^j) \cdot Q^{\text{tok}}(x^k, x^k) \\
&= \sum_{k=1}^{d}\left(\frac{1}{S} - 1\right) = \left(\frac{1}{S} - 1\right)d = Q(\mathbf{x}, \mathbf{x})
\end{aligned}
$$

for all $\mathbf{x} \in \mathcal{X}$, $i \in [d]$, and $x^i \neq \hat{x}^i \in [S]$. Then

$$
P_{s,t} = \exp((t-s)Q) = \exp\left(\bigoplus_{k=1}^{d}(t-s)Q^{\text{tok}}\right) = \left(\exp\left((t-s)Q^{\text{tok}}\right)\right)^{\otimes d} = \left(P^0_{s,t}\right)^{\otimes d}.
$$

Thus, by the Kolmogorov forward equation $\frac{\mathrm{d}}{\mathrm{d}t}q_t = Q^\top q_t$ we have

$$
q_t = \exp(tQ) \cdot p_{\text{data}} = \left(\exp\left(tQ^{\text{tok}}\right)\right)^{\otimes d} \cdot p_{\text{data}} = \left[\frac{1}{S}(1 - e^{-t})\mathbf{1}_S\mathbf{1}_S^\top + e^{-t}I_S\right]^{\otimes d} \cdot p_{\text{data}}.
$$

As $t \to +\infty$, $q_t \to \left[\frac{1}{S}\mathbf{1}_S\mathbf{1}_S^\top\right]^{\otimes d} \cdot p_{\text{data}} = \frac{1}{S^d}\mathbf{1}_{S^d}\mathbf{1}_{S^d}^\top \cdot p_{\text{data}} = \frac{1}{S^d}\mathbf{1}_{S^d} = \pi^d$. $\qquad\square$

A.2   PROOF OF PROPOSITION 2

*Proof of Proposition 2.* For the first inequality, we prove that the forward CTMC with generator matrix $Q$ and stationary distribution $\pi^d$ satisfies a modified log-Sobolev inequality (MLSI) with constant $C_{\mathrm{LSI}} = 2$:

$$\mathrm{Ent}_{\pi^d}[f] \leq \frac{C_{\mathrm{LSI}}}{2} \mathcal{E}(f, \log f) \quad \text{for all function } f : \mathcal{X} \to \mathbb{R}_+, \tag{13}$$

where the entropy is

$$\mathrm{Ent}_{\pi^d}[f] = \mathbb{E}_{\pi^d}[f \log f] - \mathbb{E}_{\pi^d}[f] \log(\mathbb{E}_{\pi^d}[f])$$

for a function $f : \mathcal{X} \to \mathbb{R}_+$, and the associated Dirichlet form is

$$\mathcal{E}(f, g) = -\int_{\mathcal{X}} f(\mathbf{x}) \sum_{\mathbf{y} \in \mathcal{X}} Q(\mathbf{x}, \mathbf{y}) g(\mathbf{y}) \, \mathrm{d}\pi^d(\mathbf{x})$$

$$= \frac{1}{2} \sum_{\mathbf{x}, \mathbf{y} \in \mathcal{X}} (f(\mathbf{x}) - f(\mathbf{y}))(g(\mathbf{x}) - g(\mathbf{y})) Q(\mathbf{x}, \mathbf{y}) \pi^d(\mathbf{x})$$

for two functions $f, g : \mathcal{X} \to \mathbb{R}$. With the sparse structure of $Q$, we can further write out that

$$\mathcal{E}(f, \log f) = \frac{1}{2S} \mathbb{E}_{\mathbf{x} \sim \pi^d} \sum_{i=1}^{d} \sum_{\hat{x}^i = 1}^{S} (f(\mathbf{x}) - f(\mathbf{x}^{\setminus i} \odot \hat{x}^i))(\log f(\mathbf{x}) - \log f(\mathbf{x}^{\setminus i} \odot \hat{x}^i))$$

for some function $f : \mathcal{X} \to \mathbb{R}_+$. We now use the subadditivity of entropy and tensorization property of log-Sobolev constants to prove (13), and we will see that it is sufficient to show that the CTMC on $[S]$ with generator matrix $Q^{\mathrm{tok}}$ and stationary distribution $\pi = \frac{1}{S} \mathbf{1}_S$ satisfies MLSI with constant $C_{\mathrm{LSI}} = 2$. First, Boucheron et al. (2013, Theorem 4.10) implies the subadditivity of entropy that

$$\mathrm{Ent}_{\pi^d}[f] \leq \mathbb{E}_{\mathbf{x}^{\setminus i} \sim \pi^{\otimes (d-1)}} \sum_{i=1}^{d} \mathrm{Ent}_{\pi}^{(i)}[f],$$

where $\mathrm{Ent}_{\pi}^{(i)}[f] = \mathbb{E}_{x^i \sim \pi}[f \log f] - \mathbb{E}_{x^i \sim \pi}[f] \log \mathbb{E}_{x^i \sim \pi}[f]$. Hence, it suffices to show that for all $i \in [d]$,

$$\mathrm{Ent}_{\pi}^{(i)}[f] \leq \frac{C_{\mathrm{LSI}}}{4} \mathbb{E}_{x^i \sim \pi} \sum_{\hat{x}^i = 1}^{S} (f(\mathbf{x}) - f(\mathbf{x}^{\setminus i} \odot \hat{x}^i))(\log f(\mathbf{x}) - \log f(\mathbf{x}^{\setminus i} \odot \hat{x}^i)). \tag{14}$$

Given any fixed realization of $\mathbf{x}^{\setminus i}$, $f(\mathbf{x})$ can take $S$ different values with equal probability $\frac{1}{S}$, and we call these values $a_1, \cdots, a_S > 0$. Then the desired inequality (14) is of the form

$$\sum_{i=1}^{S} \frac{a_i}{S} \log a_i - \left( \sum_{i=1}^{S} \frac{a_i}{S} \right) \log \sum_{i=1}^{S} \frac{a_i}{S} \leq \frac{C_{\mathrm{LSI}}}{4S^2} \sum_{i,j=1}^{S} (a_i - a_j)(\log a_i - \log a_j).$$

Thus, it remains to prove that this elementary inequality holds for all $a_1, \cdots, a_S > 0$, which can be easily verified by plugging $C_{\mathrm{LSI}} = 2$ and the concavity of logarithmic function.

To conclude, the MLSI (13) implies the exponential mixing of the forward process in KL divergence (see, e.g., Bobkov & Tetali (2006, Theorem 2.4), Chewi (2023, Theorem 1.2.25)):

$$D_{\mathrm{KL}}(q_t \| \pi^d) \leq \exp\left( -\frac{2t}{C_{\mathrm{LSI}}} \right) D_{\mathrm{KL}}(p_{\mathrm{data}} \| \pi^d) = e^{-t} D_{\mathrm{KL}}(p_{\mathrm{data}} \| \pi^d).$$

The second inequality is because

$$D_{\mathrm{KL}}(p_{\mathrm{data}} \| \pi^d) = \sum_{\mathbf{x} \in \mathcal{X}} p_{\mathrm{data}}(\mathbf{x}) \log \frac{p_{\mathrm{data}}(\mathbf{x})}{S^{-d}} = \sum_{\mathbf{x} \in \mathcal{X}} p_{\mathrm{data}}(\mathbf{x}) \log p_{\mathrm{data}}(\mathbf{x}) + d \log S \leq d \log S.$$

$\square$

### A.3 PROOF OF THEOREM 1

To prove Theorem 1, we state the following lemmas. Their proofs are provided in Appendix B.

**Lemma 1.** *The KL divergence between the true and approximate path measures of the reverse process both starting from $q_T$ is*

$$D_{\mathrm{KL}}(\mathbb{Q}\|\mathbb{P}^{q_T}) = \sum_{k=0}^{K-1} \int_{kh+\delta}^{(k+1)h+\delta} \mathbb{E}_{\mathbf{x}_t \sim q_t} \sum_{i=1}^{d} \sum_{\hat{x}_t^i \neq x_t^i} Q_t^{\mathrm{tok}}(\hat{x}_t^i, x_t^i) D_I(s_t(\mathbf{x}_t)_{i,\hat{x}_t^i} \| \hat{s}_{(k+1)h+\delta}(\mathbf{x}_t)_{i,\hat{x}_t^i}) \, \mathrm{d}t.$$

Lemma 1 is the key lemma to our analysis, allowing us to explicitly express the path measure KL divergence.

**Lemma 2** (Score bound). *Let $\delta > 0$. For all $t \in [\delta, T]$, $\mathbf{x} \in \mathcal{X}$, $i \in [d]$ and $\hat{x}^i \neq x^i \in [S]$, we have*

$$s_t(\mathbf{x})_{i,\hat{x}^i} \leq 1 + \frac{S}{e^t - 1} \leq 1 + \frac{S}{e^\delta - 1}.$$

**Lemma 3** (Score movement bound). *For all $k \in \{0, 1, \cdots, K-1\}$, $t \in [kh+\delta, (k+1)h+\delta]$, $\mathbf{x} \in \mathcal{X}$ and $\hat{x}^i \neq x^i \in [S]$, we have*

$$|s_t(\mathbf{x})_{i,\hat{x}^i} - s_{(k+1)h+\delta}(\mathbf{x})_{i,\hat{x}^i}| \lesssim \left[ \frac{e^{-(kh+\delta)}}{(1 - e^{-(kh+\delta)})^2} + \frac{S}{e^{kh+\delta} - 1} + 1 \right] Sh.$$

*Proof of Theorem 1.* Recall that $\mathbb{Q}$ is the path measure of the true reverse process $Y = (Y_t)_{t \in [0,T]}$ starting from $Y_0 \sim q_T$, and $\mathbb{P}^{\pi^d}$ is the path measure of the sampling process $Z = (Z_t)_{t \in [0,T-\delta]}$ starting from $Z_0 \sim \pi^d$. $Z$ is a CTMC with rate matrix $\hat{Q}_t^\leftarrow$, where the non-zero off-diagonal entries are defined by

$$\hat{Q}_t^\leftarrow(\mathbf{x}, \mathbf{x}^{\backslash i} \odot \hat{x}^i) = \frac{1}{S} \hat{s}_{T-[\frac{t}{h}]h}(\mathbf{x})_{i,\hat{x}^i} \quad \text{for } \hat{x}^i \neq x^i.$$

Then $\tilde{Z} = (\tilde{Z}_t)_{t \in [0,T-\delta]}$ is a CTMC with rate matrix $\hat{Q}_t^\leftarrow$ starting from $\tilde{Z}_0 \sim q_T$ with path measure $\mathbb{P}^{q_T}$. By the data processing inequality, we have the estimate

$$
\begin{aligned}
D_{\mathrm{KL}}(q_\delta \| p_{T-\delta}) \leq D_{\mathrm{KL}}(\mathbb{Q} \| \mathbb{P}^{\pi^d}) &= \mathbb{E}_{Y \sim \mathbb{Q}} \log \left( \frac{\mathrm{d}\mathbb{Q}}{\mathrm{d}\mathbb{P}^{q_T}}(Y) \frac{\mathrm{d}\mathbb{P}^{q_T}}{\mathrm{d}\mathbb{P}^{\pi^d}}(Y) \right) \\
&= \mathbb{E}_{Y \sim \mathbb{Q}} \log \left( \frac{\mathrm{d}\mathbb{Q}}{\mathrm{d}\mathbb{P}^{q_T}}(Y) \frac{\mathrm{d}q_T}{\mathrm{d}\pi^d}(Y_0) \right) \\
&= D_{\mathrm{KL}}(\mathbb{Q} \| \mathbb{P}^{q_T}) + D_{\mathrm{KL}}(q_T \| \pi^d),
\end{aligned}
\tag{15}
$$

where the second to last equality holds since the only difference between the two path measures $\mathbb{P}^{q_T}$ and $\mathbb{P}^{\pi^d}$ is the initial distribution for a path $Y$. We respectively bound the two terms in (15). The second term is bounded by Proposition 2. We next combine Lemmas 1, 2, and 3, Assumption 1 and Proposition 3 to bound the first term in (15). By choosing

$$C_1 = \max \left\{ 1 + \frac{S}{e^\delta - 1}, \max_{\mathbf{x} \in \mathcal{X}, k \in \{0, \cdots, K-1\}} \|\hat{s}_{(k+1)h+\delta}(\mathbf{x})\|_\infty \right\},$$

we have

$$
\begin{aligned}
&D_{\mathrm{KL}}(\mathbb{Q} \| \mathbb{P}^{q_T}) \\
&= \frac{1}{S} \sum_{k=0}^{K-1} \int_{kh+\delta}^{(k+1)h+\delta} \mathbb{E}_{\mathbf{x}_t \sim q_t} D_I(s_t(\mathbf{x}_t) \| \hat{s}_{(k+1)h+\delta}(\mathbf{x}_t)) \, \mathrm{d}t \\
&\lesssim \frac{C_1}{S} \sum_{k=0}^{K-1} \int_{kh+\delta}^{(k+1)h+\delta} \mathbb{E}_{\mathbf{x}_t \sim q_t} \|s_t(\mathbf{x}_t) - s_{(k+1)h+\delta}(\mathbf{x}_t)\|_2^2 \, \mathrm{d}t \\
&\quad + \frac{C_1^2}{S} \sum_{k=0}^{K-1} \int_{kh+\delta}^{(k+1)h+\delta} \mathbb{E}_{\mathbf{x}_t \sim q_t} D_I(s_{(k+1)h+\delta}(\mathbf{x}_t) \| \hat{s}_{(k+1)h+\delta}(\mathbf{x}_t)) \, \mathrm{d}t
\end{aligned}
$$

$$\lesssim \frac{C_1}{S} \sum_{k=0}^{K-1} \left[ \frac{e^{-(kh+\delta)}}{(1-e^{-(kh+\delta)})^2} + \frac{S}{e^{kh+\delta}-1} + 1 \right]^2 dS^3 h^3 + C_1^2 \epsilon_{\text{score}}$$

$$\lesssim C_1 \sum_{k=0}^{K-1} \left[ \frac{e^{-2(kh+\delta)}}{(1-e^{-(kh+\delta)})^4} + \left( \frac{S}{e^{kh+\delta}-1} \right)^2 + 1 \right] dS^2 h^3 + C_1^2 \epsilon_{\text{score}}$$

$$\lesssim C_1 dS^2 h^3 \int_\delta^T \left[ \frac{e^{-2x}}{(1-e^{-x})^4} + \frac{S^2}{e^x-1} \right] \mathrm{d}x + C_1 dS^2 h^2 T + C_1^2 \epsilon_{\text{score}}$$

$$\leq \left[ (e^\delta-1)^{-3} + (\delta - \log(e^\delta-1))S^2 \right] C_1 dS^2 h^3 + C_1 dS^2 h^2 T + C_1^2 \epsilon_{\text{score}}$$

$$\leq \left[ \delta^{-3} + (\delta + \log(1/\delta))S^2 \right] C_1 dS^2 h^3 + C_1 dS^2 h^2 T + C_1^2 \epsilon_{\text{score}}$$

$$\lesssim \delta^{-3} C_1 dS^2 h^3 + C_1 dS^2 h^2 T + C_1^2 \epsilon_{\text{score}}. \tag{16}$$

The first equality follows from Lemma 1. The first inequality is a consequence of Lemma 2 and Proposition 3. Lemma 3 and Assumption 1 lead to the second inequality. The third inequality stems from $(a+b+c)^2 \leq 3(a^2+b^2+c^2)$. The second last inequality utilizes the fact that $e^x - 1 \geq x$. For the final inequality, we employ $\delta = \tilde{O}(S^{-\frac{2}{3}})$ and note that $\delta^{-3} + \log(1/\delta)S^2 = O(\delta^{-3})$ as $\delta \to 0^+$. Lastly, combining (15), (16), and Proposition 2 yields the KL divergence bound (8).

As for the TV distance bound (9), it is a similar proof to that of Theorem 6(2) in Chen & Ying (2024). By the definition of TV distance and the uniformization of CTMC (Chen & Ying, 2024, Proposition 1), we have

$$\begin{aligned} D_{\text{TV}}(p_{\text{data}}, q_\delta) &\leq \mathbb{P}(X_0 \neq X_\delta) \\ &\leq \mathbb{P}(\text{'A Poisson}(d(S-1)\delta/S) \text{ random variable is non-zero'}) \\ &= 1 - e^{-d(S-1)\delta/S}. \end{aligned} \tag{17}$$

Then by the triangle inequality, Pinsker's inequality and inequality (8), we have

$$\begin{aligned} &D_{\text{TV}}(p_{\text{data}}, p_{T-\delta}) \\ &\leq D_{\text{TV}}(q_\delta, p_{T-\delta}) + D_{\text{TV}}(p_{\text{data}}, q_\delta) \\ &\leq \sqrt{\frac{1}{2} D_{\text{KL}}(q_\delta \| p_{T-\delta})} + (1 - e^{-d(S-1)\delta/S}) \\ &\lesssim \sqrt{de^{-T}\log S + \delta^{-3}C_1 S^2 h^3 d + C_1 S^2 h^2 dT + C_1^2 \epsilon_{\text{score}}} + (1 - e^{-d(S-1)\delta/S}). \end{aligned}$$

We conclude the proof of Theorem 1. $\qquad\square$

### A.4 PROOF OF THEOREM 2

*Proof of Theorem 2.* The proof of Theorem 2 is similar to that of Theorem 1. Note that the decomposition (15) and the equation for path measure KL divergence in Lemma 1 still hold for $\delta = 0$. It suffices to bound the first term of (15). We need the following lemmas. Proofs of Lemmas 4 and 5 are provided in Appendix B.

**Lemma 4** (Score bound). *Suppose Assumption 2 holds. Let $\delta = 0$. Then for all $t \in [0, T]$, $\mathbf{x} \in \mathcal{X}$, $i \in [d]$ and $\hat{x}^i \neq x^i \in [S]$, we have*

$$s_t(\mathbf{x})_{i,\hat{x}^i} \leq L.$$

**Lemma 5** (Score movement bound). *Suppose Assumption 2 holds. Let $\delta = 0$. Then for all $k \in \{0, 1, \cdots, K-1\}$, $t \in [kh, (k+1)h]$, $\mathbf{x} \in \mathcal{X}$, $i \in [d]$, and $\hat{x}^i \neq x^i \in [S]$, we have*

$$|s_t(\mathbf{x})_{i,\hat{x}^i} - s_{(k+1)h}(\mathbf{x})_{i,\hat{x}^i}| \lesssim \left[ \frac{1}{1-e^{-(k+1)h}} + S \right] \kappa_i h.$$

We then combine Lemmas 1, 4, and 5, Assumption 1 and Proposition 3 with $\delta = 0$. By choosing

$$C_2 = \max \left\{ L, \max_{\mathbf{x}\in\mathcal{X}, k\in\{0,\cdots,K-1\}} \|\hat{s}_{(k+1)h}(\mathbf{x})\|_\infty \right\},$$

we have

$$D_{\mathrm{KL}}(\mathbb{Q}\|\mathbb{P}^{q_T})$$

$$= \frac{1}{S}\sum_{k=0}^{K-1}\int_{kh}^{(k+1)h}\mathbb{E}_{\mathbf{x}_t\sim q_t}D_I(s_t(\mathbf{x}_t)\|\hat{s}_{(k+1)h}(\mathbf{x}_t))\,\mathrm{d}t$$

$$\lesssim \frac{C_2}{S}\sum_{k=0}^{K-1}\int_{kh}^{(k+1)h}\mathbb{E}_{\mathbf{x}_t\sim q_t}\|s_t(\mathbf{x}_t)-s_{(k+1)h}(\mathbf{x}_t)\|_2^2\,\mathrm{d}t$$

$$+\frac{C_2^2}{S}\sum_{k=0}^{K-1}\int_{kh}^{(k+1)h}\mathbb{E}_{\mathbf{x}_t\sim q_t}D_I(s_{(k+1)h}(\mathbf{x}_t)\|\hat{s}_{(k+1)h}(\mathbf{x}_t))\,\mathrm{d}t$$

$$\leq \frac{C_2}{S}\sum_{k=0}^{K-1}\int_{kh}^{(k+1)h}\mathbb{E}_{\mathbf{x}_t\sim q_t}\sum_{i=1}^{d}\sum_{\hat{x}^i\neq x^i}|s_t(\mathbf{x}_t)_{i,\hat{x}^i}-s_{(k+1)h}(\mathbf{x}_t)_{i,\hat{x}^i}|^2\,\mathrm{d}t+C_2^2\epsilon_{\mathrm{score}}$$

$$\lesssim \frac{C_2}{S}\sum_{k=0}^{K-1}\int_{kh}^{(k+1)h}\mathbb{E}_{\mathbf{x}_t\sim q_t}\sum_{i=1}^{d}\sum_{\hat{x}^i\neq x^i}\left[\frac{1}{1-e^{-(k+1)h}}+S\right]^2\kappa_i^2h^2\,\mathrm{d}t+C_2^2\epsilon_{\mathrm{score}}$$

$$\lesssim C_2\sum_{k=0}^{K-1}\sum_{i=1}^{d}\left[\frac{1}{(1-e^{-(k+1)h})^2}+S^2\right]\kappa_i^2h^3+C_2^2\epsilon_{\mathrm{score}}$$

$$= C_2\kappa^2\sum_{k=0}^{K-1}\frac{1}{(1-e^{-(k+1)h})^2}h^3+C_2\kappa^2S^2h^2T+C_2^2\epsilon_{\mathrm{score}}$$

$$\lesssim C_2\kappa^2h^3\int_{h}^{T}\frac{1}{(1-e^{-x})^2}\,\mathrm{d}x+C_2\kappa^2S^2h^2T+C_2^2\epsilon_{\mathrm{score}}$$

$$\leq C_2\kappa^2h^3\left[T-\log(e^h-1)+\frac{1}{e^h-1}\right]+C_2\kappa^2S^2h^2T+C_2^2\epsilon_{\mathrm{score}}$$

$$\overset{(i)}{\leq} C_2\kappa^2h^3\left[T-\log h+\frac{1}{h}\right]+C_2\kappa^2S^2h^2T+C_2^2\epsilon_{\mathrm{score}}$$

$$\lesssim C_2\kappa^2h^3\left[T+\frac{1}{h}\right]+C_2\kappa^2S^2h^2T+C_2^2\epsilon_{\mathrm{score}}$$

$$\lesssim C_2\kappa^2h^2T\left[h+S^2\right]+C_2^2\epsilon_{\mathrm{score}}$$

$$\overset{(ii)}{\lesssim} C_2\kappa^2h^2S^2T+C_2^2\epsilon_{\mathrm{score}}. \tag{18}$$

The first equality follows from Lemma 1. Lemma 4 and Proposition 3 lead to the first inequality. Assumption 1 gives rise to the second inequality, while Lemma 5 yields the third. The fourth inequality stems from the fact that $(a+b)^2\leq 2(a^2+b^2)$. Inequality $(i)$ is a consequence of $e^x-1\geq x$, and $(ii)$ results from the assumption that $h\leq S^2$.

Then Theorem 2 follows from (15), (18) and Proposition 2. $\qquad\square$

**Discussion on $C_1$ and $C_2$.** Since we have access to uniform bounds for the true score functions from Lemmas 2 and 4, which depend on either $\delta$ and $S$, or $L$, we can apply score clipping during training to ensure that the learned score functions are reliable. Specifically, we enforce the following conditions:

$$\max_{\mathbf{x}\in\mathcal{X},\,k\in\{0,\ldots,K-1\}}\|\hat{s}_{(k+1)h+\delta}(\mathbf{x})\|_\infty\leq\frac{3}{2}\left(1+\frac{S}{e^\delta-1}\right)\leq\frac{3S}{\delta},\quad\text{for a small }\delta.$$

and

$$\max_{\mathbf{x}\in\mathcal{X},\,k\in\{0,\ldots,K-1\}}\|\hat{s}_{(k+1)h}(\mathbf{x})\|_\infty\leq\frac{3}{2}L.$$

As a result, we can choose that $C_1=\frac{3S}{\delta}$ and $C_2=\frac{3}{2}L$.

## B    OMITTED PROOFS IN APPENDIX A

### B.1    PROOF OF LEMMA 1

To prove Lemma 1, we need the following generalized Girsanov's theorem and Dynkin's formula, as stated below. For related definitions and details regarding the notations used in Theorem 3 and Lemma 6, we refer readers to the work of Benton et al. (2024b).

**Theorem 3** (Girsanov, (Benton et al., 2024b, Theorem 6)). *Let $\bar{Y} = (Y_t, t)_{t \geq 0}$ and $\bar{Z} = (Z_t, t)_{t \geq 0}$ be Feller processes on $\mathcal{S}$ with generators $\mathcal{L}, \mathcal{M}$ and path measures $\bar{\mathbb{Q}}, \bar{\mathbb{P}}$ respectively, such that $Y_0$ and $Z_0$ have the same law. Suppose that there exists a bounded, measurable function $\alpha : \mathcal{S} \rightarrow (0, +\infty)$ such that $\alpha^{-1} \mathcal{L} \alpha$ is bounded, and such that*

$$\alpha \mathcal{M} f = \mathcal{L}(f\alpha) - f \mathcal{L} \alpha \qquad (19)$$

*for all proper functions $f$ (assume that all the involved functions are well-defined). Then we have*

$$\frac{\mathrm{d}\bar{\mathbb{P}}}{\mathrm{d}\bar{\mathbb{Q}}}(\omega) = \frac{\alpha(\omega_T, T)}{\alpha(\omega_0, 0)} \exp\left\{ -\int_0^T \frac{\mathcal{L}\alpha(\omega_s, s)}{\alpha(\omega_s, s)} \, \mathrm{d}s \right\}.$$

**Lemma 6** (Dynkin's formula, (Dong, 2003, Theorem 27.20)). *If $\bar{Y} = (Y_t, t)_{t \geq 0}$ is a Feller process on $\mathcal{S}$ with generator $\mathcal{L}$ and $f$ is a proper function on $\mathcal{S}$ (assume that all the involved functions are well-defined), then*

$$M_t^f = f(Y_t, t) - f(Y_0, 0) - \int_0^t \mathcal{L}f(Y_s, s) \, \mathrm{d}s$$

*is a martingale with respect to the natural filtration of $\bar{Y}$.*

*Proof of Lemma 1.* Recall that $Y = (Y_t)_{t \in [0,T]}$ is the true time reversal CTMC with generator $Q_t^{\leftarrow}$, and $Z = (Z_t)_{t \in [0,T]}$ is the sampling CTMC with generator $\hat{Q}_t^{\leftarrow}$, both initiating from $q_T$; the rate matrices are of the forms

$$Q_t^{\leftarrow}(\mathbf{x}, \mathbf{x}^{\backslash i} \odot \hat{x}^i) = \frac{1}{S} s_{T-t}(\mathbf{x})_{i,\hat{x}^i} \quad \text{and} \quad \hat{Q}_t^{\leftarrow}(\mathbf{x}, \mathbf{x}^{\backslash i} \odot \hat{x}^i) = \frac{1}{S} \hat{s}_{T-[\frac{t}{h}]h}(\mathbf{x})_{i,\hat{x}^i} \quad \text{for } x^i \neq \hat{x}^i.$$

Since the processes $Y$ and $Z$ on $\mathcal{X}$ are time-inhomogeneous, inspired by Benton et al. (2024b), we consider Feller processes $\bar{Y}, \bar{Z}$ defined on the extended space $\mathcal{S} = \mathcal{X} \times [0, +\infty)$ which are constructed by setting $Y_t = Y_T$ and $Z_t = Z_T$ for $t \geq T$ and letting $\bar{Y} = (Y_t, t)_{t \geq 0}$ and $\bar{Z} = (Z_t, t)_{t \geq 0}$, which are time-homogeneous. For more related details on the Feller process and stochastic analysis, we refer readers to the work of Benton et al. (2024b).

We can now apply Theorem 3 with $\mathcal{L} = \partial_t + \hat{\mathcal{L}}$ and $\mathcal{M} = \partial_t + \hat{\mathcal{M}}$, where $\hat{\mathcal{L}}$ and $\hat{\mathcal{M}}$ are the generators of the CTMCs $Y$ and $Z$ respectively, the condition (19) has the form

$$\alpha \hat{\mathcal{M}} f = \hat{\mathcal{L}}(f\alpha) - f \hat{\mathcal{L}} \alpha,$$

and it follows that

$$\alpha(\mathbf{x}, t) \sum_{\mathbf{z} \in \mathcal{X}} \hat{Q}_t^{\leftarrow}(\mathbf{x}, \mathbf{z}) f(\mathbf{z}) = \sum_{\mathbf{z} \in \mathcal{X}} Q_t^{\leftarrow}(\mathbf{x}, \mathbf{z}) \alpha(\mathbf{z}, t) f(\mathbf{z}) - f(\mathbf{x}) \sum_{\mathbf{z} \in \mathcal{X}} Q_t^{\leftarrow}(\mathbf{x}, \mathbf{z}) \alpha(\mathbf{z}, t)$$

for all $\mathbf{x} \in \mathcal{X}$. By taking $\mathbf{y} = \mathbf{x}^{\backslash i} \odot \hat{x}^i$ $(\hat{x}^i \neq x^i)$ and $f(\mathbf{z}) = \mathbf{1}\{\mathbf{z} = \mathbf{y}\}$, we have

$$\alpha(\mathbf{x}, t) \hat{s}_{T-[\frac{t}{h}]h}(\mathbf{x})_{i,\hat{x}^i} = \alpha(\mathbf{y}, t) s_{T-t}(\mathbf{x})_{i,\hat{x}^i} \quad \text{for all } i \in [d] \text{ and } \hat{x}^i \neq x^i.$$

Thus, in order for (19) to hold, it is required that

$$s_{T-t}(\mathbf{x})_{i,\hat{x}^i} = \frac{\alpha(\mathbf{x}, t)}{\alpha(\mathbf{x}^{\backslash i} \odot \hat{x}^i, t)} \hat{s}_{T-[\frac{t}{h}]h}(\mathbf{x})_{i,\hat{x}^i} \quad \text{for all } i \in [d] \text{ and } \hat{x}^i \neq x^i. \qquad (20)$$

It also can be easily verified that this is sufficient for (19) to hold for a given $\alpha$. Let $\bar{\mathbb{Q}}$ and $\bar{\mathbb{P}}$ be the path measures of $\bar{Y}$ and $\bar{Z}$ respectively. Then with function $\alpha(\mathbf{x}, t)$ satisfying (20), Theorem 3 yields that

$$\frac{\mathrm{d}\bar{\mathbb{P}}}{\mathrm{d}\bar{\mathbb{Q}}}(Y) = \frac{\alpha(Y_{T-\delta}, T-\delta)}{\alpha(Y_0, 0)} \exp\left\{ -\int_0^{T-\delta} \alpha(Y_s, s)^{-1} \mathcal{L}\alpha(Y_s, s) \, \mathrm{d}s \right\}. \qquad (21)$$

By taking $f = \log \alpha$ in Lemma 6, we know that

$$M_t^f := \log \frac{\alpha(Y_t, t)}{\alpha(Y_0, 0)} - \int_0^t \mathcal{L} \log \alpha(Y_s, s)\, \mathrm{d}s$$

is a $\bar{\mathbb{Q}}$-martingale. Then by taking logarithms in (21), it follows that

$$\log \frac{\mathrm{d}\bar{\mathbb{P}}}{\mathrm{d}\bar{\mathbb{Q}}}(Y)$$

$$= \int_0^{T-\delta} \left[ \mathcal{L} \log \alpha(Y_s, s) - \frac{\mathcal{L}\alpha(Y_s, s)}{\alpha(Y_s, s)} \right] \mathrm{d}s + M_{T-\delta}^f$$

$$= \int_0^{T-\delta} \left[ \left[ \partial_t \log \alpha(Y_s, s) + \hat{\mathcal{L}} \log \alpha(Y_s, s) \right] - \left[ \frac{\partial_t \alpha(Y_s, s)}{\alpha(Y_s, s)} + \frac{\hat{\mathcal{L}}\alpha(Y_s, s)}{\alpha(Y_s, s)} \right] \right] \mathrm{d}s + M_{T-\delta}^f$$

$$= \int_0^{T-\delta} \left[ \hat{\mathcal{L}} \log \alpha(Y_s, s) - \frac{\hat{\mathcal{L}}\alpha(Y_s, s)}{\alpha(Y_s, s)} \right] \mathrm{d}s + M_{T-\delta}^f.$$

Taking expectation above yields that

$$D_{\mathrm{KL}}(\mathbb{Q}\|\mathbb{P}^{q_T}) = D_{\mathrm{KL}}(\bar{\mathbb{Q}}\|\bar{\mathbb{P}}) = \mathbb{E}_{\bar{\mathbb{Q}}} \log \frac{\mathrm{d}\bar{\mathbb{Q}}}{\mathrm{d}\bar{\mathbb{P}}}$$

$$= \mathbb{E}_{\bar{\mathbb{Q}}} \int_0^{T-\delta} \left[ \frac{\hat{\mathcal{L}}\alpha(Y_s, s)}{\alpha(Y_s, s)} - \hat{\mathcal{L}} \log \alpha(Y_s, s) \right] \mathrm{d}s + \mathbb{E}_{\bar{\mathbb{Q}}} M_{T-\delta}^f$$

$$= \int_0^{T-\delta} \mathbb{E}_{\mathbf{x}_t \sim q_{T-t}} \left[ \frac{\hat{\mathcal{L}}\alpha(\mathbf{x}_t, t)}{\alpha(\mathbf{x}_t, t)} - \hat{\mathcal{L}} \log \alpha(\mathbf{x}_t, t) \right] \mathrm{d}t$$

$$= \int_0^{T-\delta} \mathbb{E}_{\mathbf{x}_t \sim q_{T-t}} \left[ \sum_{\mathbf{y} \in \mathcal{X}} \left\{ Q_t^{\leftarrow}(\mathbf{x}_t, \mathbf{y}) \frac{\alpha(\mathbf{y}, t)}{\alpha(\mathbf{x}_t, t)} - Q_t^{\leftarrow}(\mathbf{x}_t, \mathbf{y}) \log \frac{\alpha(\mathbf{y}, t)}{\alpha(\mathbf{x}_t, t)} \right\} \right] \mathrm{d}t$$

$$= \int_0^{T-\delta} \mathbb{E}_{\mathbf{x}_t \sim q_{T-t}} \left[ Q_t^{\leftarrow}(\mathbf{x}_t, \mathbf{x}_t) + \sum_{\mathbf{y} \neq \mathbf{x}_t} Q_t^{\leftarrow}(\mathbf{x}_t, \mathbf{y}) \frac{\alpha(\mathbf{y}, t)}{\alpha(\mathbf{x}_t, t)} + \sum_{\mathbf{y} \neq \mathbf{x}_t} Q_t^{\leftarrow}(\mathbf{x}_t, \mathbf{y}) \log \frac{\alpha(\mathbf{x}_t, t)}{\alpha(\mathbf{y}, t)} \right] \mathrm{d}t$$

$$= \frac{1}{S} \int_0^{T-\delta} \mathbb{E}_{\mathbf{x}_t \sim q_{T-t}} \sum_{i=1}^{d} \sum_{\hat{x}_t^i \neq x_t^i} \left[ -s_{T-t}(\mathbf{x}_t)_{i,\hat{x}_t^i} + \hat{s}_{T-[\frac{t}{h}]h}(\mathbf{x}_t)_{i,\hat{x}_t^i} \right.$$

$$\left. + s_{T-t}(\mathbf{x}_t)_{i,\hat{x}_t^i} \log \frac{s_{T-t}(\mathbf{x}_t)_{i,\hat{x}_t^i}}{\hat{s}_{T-[\frac{t}{h}]h}(\mathbf{x}_t)_{i,\hat{x}_t^i}} \right] \mathrm{d}t$$

$$= \frac{1}{S} \int_0^{T-\delta} \mathbb{E}_{\mathbf{x}_t \sim q_{T-t}} \sum_{i=1}^{d} \sum_{\hat{x}_t^i \neq x_t^i} D_I\left(s_{T-t}(\mathbf{x}_t)_{i,\hat{x}_t^i} \| \hat{s}_{T-[\frac{t}{h}]h}(\mathbf{x}_t)_{i,\hat{x}_t^i}\right) \mathrm{d}t$$

$$= \frac{1}{S} \sum_{k=0}^{K-1} \int_{kh}^{(k+1)h} \mathbb{E}_{\mathbf{x}_t \sim q_{T-t}} \sum_{i=1}^{d} \sum_{\hat{x}_t^i \neq x_t^i} D_I\left(s_{T-t}(\mathbf{x}_t)_{i,\hat{x}_t^i} \| \hat{s}_{T-kh}(\mathbf{x}_t)_{i,\hat{x}_t^i}\right) \mathrm{d}t$$

$$= \frac{1}{S} \sum_{k=0}^{K-1} \int_{kh}^{(k+1)h} \mathbb{E}_{\mathbf{x}_t \sim q_{T-t}} D_I\left(s_{T-t}(\mathbf{x}_t) \| \hat{s}_{T-kh}(\mathbf{x}_t)\right) \mathrm{d}t$$

$$= \frac{1}{S} \sum_{k=0}^{K-1} \int_{kh+\delta}^{(k+1)h+\delta} \mathbb{E}_{\mathbf{x}_t \sim q_t} D_I\left(s_t(\mathbf{x}_t) \| \hat{s}_{(k+1)h+\delta}(\mathbf{x}_t)\right) \mathrm{d}t,$$

where we used $\mathbb{E}_{\bar{\mathbb{Q}}} M_{T-\delta}^f = \mathbb{E}_{\bar{\mathbb{Q}}} M_0^f = 0$ since $\{M_t^f\}_{t \geq 0}$ is a $\bar{\mathbb{Q}}$-martingale by Lemma 6. Thus we finish the proof of Lemma 1. $\qquad\square$

## B.2    PROOF OF LEMMA 2

*Proof o Lemma 2.* Note that for all $0 \leq t' \neq t$,

$$
\begin{aligned}
s_t(\mathbf{x})_{i,\hat{x}^i} = \frac{q_t(\mathbf{x}^{\backslash i} \odot \hat{x}^i)}{q_t(\mathbf{x})} &= \frac{1}{q_t(\mathbf{x})} \sum_{\mathbf{x}_{t'} \in \mathcal{X}} q_{t'}(\mathbf{x}_{t'}) q_{t|t'}(\mathbf{x}^{\backslash i} \odot \hat{x}^i | \mathbf{x}_{t'}) \\
&= \frac{1}{q_t(\mathbf{x})} \sum_{\mathbf{x}_{t'} \in \mathcal{X}} q_{t'}(\mathbf{x}_{t'}) \prod_{k \neq i} q_{t|t'}^i(x^k | x_{t'}^k) q_{t|t'}^i(\hat{x}^i | x_{t'}^i) \\
&= \frac{1}{q_t(\mathbf{x})} \sum_{\mathbf{x}_{t'} \in \mathcal{X}} q_{t'}(\mathbf{x}_{t'}) \prod_{k=1}^d q_{t|t'}^i(x^k | x_{t'}^k) \frac{q_{t|t'}^i(\hat{x}^i | x_{t'}^i)}{q_{t|t'}^i(x^i | x_{t'}^i)} \\
&= \sum_{\mathbf{x}_{t'} \in \mathcal{X}} \frac{q_{t'}(\mathbf{x}_{t'}) q_{t|t'}(\mathbf{x} | \mathbf{x}_{t'})}{q_t(\mathbf{x})} \cdot \frac{q_{t|t'}^i(\hat{x}^i | x_{t'}^i)}{q_{t|t'}^i(x^i | x_{t'}^i)} \\
&= \mathbb{E}_{\mathbf{x}_{t'} \sim q_{t'|t}(\cdot | \mathbf{x})} \frac{q_{t|t'}^i(\hat{x}^i | x_{t'}^i)}{q_{t|t'}^i(x^i | x_{t'}^i)}.
\end{aligned}
\tag{22}
$$

In particular, by taking $t' = 0$ in (22), we have

$$
\begin{aligned}
s_t(\mathbf{x})_{i,\hat{x}^i} &= \mathbb{E}_{\mathbf{x}_0 \sim q_{0|t}(\cdot|\mathbf{x})} \frac{q_{t|0}^i(\hat{x}^i | x_0^i)}{q_{t|0}^i(x^i | x_0^i)} \\
&= \mathbb{E}_{\mathbf{x}_0 \sim q_{0|t}(\cdot|\mathbf{x})} \frac{1 + e^{-t}(-1 + S \cdot \delta\{\hat{x}^i, x_0^i\})}{1 + e^{-t}(-1 + S \cdot \delta\{x^i, x_0^i\})} \leq 1 + \frac{S}{e^t - 1} \leq 1 + \frac{S}{e^\delta - 1}.
\end{aligned}
\tag{23}
$$

$\square$

## B.3    PROOF OF LEMMA 3

To prove Lemma 3, we need the following lemmas. Proofs of Lemmas 7, 8, and 9 are provided in Appendix C.

**Lemma 7.** *Denote $q_t^i$ as the marginals of the $i$-th dimensional forward CTMC $X^i$, and the corresponding score function for $X^i$ as $s_t^i(x)_y = \frac{q_t^i(y)}{q_t^i(x)}$ for $x \neq y \in [S]$. Then for all $i \in [d]$, $k \in \{0, 1, \cdots, K-1\}$, $t \in [kh + \delta, (k+1)h + \delta]$, and $x \neq y \in [S]$, we have*

$$
s_t^i(x)_y \leq 1 + \frac{S}{e^{kh+\delta} - 1}, \quad \text{and} \quad \left| s_t^i(x)_y - s_{(k+1)h+\delta}^i(x)_y \right| \leq \frac{She^{-(kh+\delta)}}{(1 - e^{-(kh+\delta)})^2}.
$$

**Lemma 8.** *There exists some large enough $t'$ such that for all $i \in [d]$, $k \in \{0, 1, \cdots, K-1\}$, $t \in [kh + \delta, (k+1)h + \delta]$, $x \neq y \in [S]$, and $x_{t'} \in [S]$, we have*

$$
\left| \frac{q_{t'|t}^i(x_{t'}|y)}{q_{t'|t}^i(x_{t'}|x)} - \frac{q_{t'|(k+1)h+\delta}^i(x_{t'}|y)}{q_{t'|(k+1)h+\delta}^i(x_{t'}|x)} \right| \leq 8Sh, \quad \text{and} \quad \frac{q_{t'|t}^i(x_{t'}|y)}{q_{t'|t}^i(x_{t'}|x)} \leq 2.
$$

**Lemma 9.** *There exists some large enough $t'$ such that for all $k \in \{0, 1, \cdots, K-1\}$, $t \in [kh + \delta, (k+1)h + \delta]$ and $\mathbf{x} \in \mathcal{X}$, it holds that*

$$
2 \cdot D_{\text{TV}}(q_{t'|t}(\cdot|\mathbf{x}), q_{t'|(k+1)h+\delta}(\cdot|\mathbf{x})) \leq h.
$$

*Proof of Lemma 3.* We can bound the high-dimensional score movement as follows:

$$
\begin{aligned}
&\left| s_t(\mathbf{x})_{i,\hat{x}^i} - s_{(k+1)h+\delta}(\mathbf{x})_{i,\hat{x}^i} \right| \\
&= \left| \mathbb{E}_{\mathbf{x}_{t'} \sim q_{t'|t}(\cdot|\mathbf{x})} \frac{q_{t|t'}^i(\hat{x}^i | x_{t'}^i)}{q_{t|t'}^i(x^i | x_{t'}^i)} - \mathbb{E}_{\mathbf{x}_{t'} \sim q_{t'|(k+1)h+\delta}(\cdot|\mathbf{x})} \frac{q_{(k+1)h+\delta|t'}^i(\hat{x}^i | x_{t'}^i)}{q_{(k+1)h+\delta|t'}^i(x^i | x_{t'}^i)} \right| \\
&\leq \mathbb{E}_{\mathbf{x}_{t'} \sim q_{t'|t}(\cdot|\mathbf{x})} \left| \frac{q_{t|t'}^i(\hat{x}^i | x_{t'}^i)}{q_{t|t'}^i(x^i | x_{t'}^i)} - \frac{q_{(k+1)h+\delta|t'}^i(\hat{x}^i | x_{t'}^i)}{q_{(k+1)h+\delta|t'}^i(x^i | x_{t'}^i)} \right|
\end{aligned}
$$

$$+ \left| \mathbb{E}_{\mathbf{x}_{t'} \sim q_{t'|t}(\cdot|\mathbf{x})} \frac{q^i_{(k+1)h+\delta|t'}(\hat{x}^i|x^i_{t'})}{q^i_{(k+1)h+\delta|t'}(x^i|x^i_{t'})} - \mathbb{E}_{\mathbf{x}_{t'} \sim q_{t'|(k+1)h+\delta}(\cdot|\mathbf{x})} \frac{q^i_{(k+1)h+\delta|t'}(\hat{x}^i|x^i_{t'})}{q^i_{(k+1)h+\delta|t'}(x^i|x^i_{t'})} \right|$$

$$\leq \mathbb{E}_{\mathbf{x}_{t'} \sim q_{t'|t}(\cdot|\mathbf{x})} \left| s_t(x^i)_{\hat{x}^i} \cdot \frac{q^i_{t'|t}(x^i_{t'}|\hat{x}^i)}{q^i_{t'|t}(x^i_{t'}|x^i)} - s_{(k+1)h+\delta}(x^i)_{\hat{x}^i} \cdot \frac{q^i_{t'|(k+1)h+\delta}(x^i_{t'}|\hat{x}^i)}{q^i_{t'|(k+1)h+\delta}(x^i_{t'}|x^i)} \right|$$

$$+ \sup_{t,\hat{x}^i,x^i_{t'},x^i} \left\{ \frac{q^i_{t'|t}(x^i_{t'}|\hat{x}^i)}{q^i_{t'|t}(x^i_{t'}|x^i)} \right\} \cdot 2 D_{\mathrm{TV}}(q_{t'|t}(\cdot|\mathbf{x}), q_{t'|(k+1)h+\delta}(\cdot|\mathbf{x})), \tag{24}$$

where the first equality is by (22), the first inequality is due to the triangle inequality, and the last inequality is due to the Bayes' rule.

Then by utilizing the inequality $|a_1 a_2 - b_1 b_2| \leq |a_1 - b_1|a_2 + b_1|a_2 - b_2|$ $(a_1, a_2, b_1, b_2 \geq 0)$ for the first term in (24), and then using Lemmas 7, 8, and 9, for some large enough $t'$, we have that

$$|s_t(\mathbf{x})_{i,\hat{x}^i} - s_{(k+1)h+\delta}(\mathbf{x})_{i,\hat{x}^i}| \leq \left[ \frac{2She^{-(kh+\delta)}}{(1 - e^{-(kh+\delta)})^2} + (1 + \frac{S}{e^{kh+\delta} - 1})8Sh \right] + 2h$$

$$\lesssim \left[ \frac{e^{-(kh+\delta)}}{(1 - e^{-(kh+\delta)})^2} + \frac{S}{e^{kh+\delta} - 1} + 1 \right] Sh.$$

$\square$

## B.4 PROOF OF LEMMA 4

*Proof of Lemma 4.* This is a similar proof to that of Lemma 8 in Chen & Ying (2024). Define the kernel function

$$g_{\mathbf{w}}(t) = \frac{1}{S^d} \prod_{i=1}^d \left[ 1 + e^{-t}(-1 + S \cdot \mathbf{1}\{w^i \equiv 0 \pmod{S}\}) \right] \quad \text{for } \mathbf{w} \in \mathbb{Z}^d, \ t \geq 0.$$

By Proposition 1, we can express the transition probability of the forward process as

$$q_{t|s}(\mathbf{x}_t|\mathbf{x}_s) = \prod_{i=1}^d q^i_{t|s}(x^i_t|x^i_s) = \prod_{i=1}^d P^0_{s,t}(x^i_s, x^i_t)$$

$$= \frac{1}{S^d} \prod_{i=1}^d \left[ 1 + e^{-(t-s)}(-1 + S \cdot \delta\{x^i_t, x^i_s\}) \right] \quad \text{for all } t > s \geq 0.$$

Then we have $q_{t|0}(\mathbf{y}|\mathbf{x}) = g_{\mathbf{y}-\mathbf{x}}(t)$ for $t > 0$. Thus, it follows that for all $t \in (0, T]$, $\mathbf{x} \in \mathcal{X}$, $i \in [d]$ and $x^i \neq \hat{x}^i \in [S]$,

$$s_t(\mathbf{x})_{i,\hat{x}^i} = \frac{q_t(\mathbf{x}^{\backslash i} \odot \hat{x}^i)}{q_t(\mathbf{x})}$$

$$= \frac{q_t(\mathbf{x} + (\hat{x}^i - x^i)\mathbf{e}_i)}{q_t(\mathbf{x})}$$

$$= \frac{\sum_{\mathbf{y} \in \mathcal{X}} q_0(\mathbf{y})q_{t|0}(\mathbf{x} + (\hat{x}^i - x^i)\mathbf{e}_i|\mathbf{y})}{q_t(\mathbf{x})}$$

$$= \frac{\sum_{\mathbf{y} \in \mathcal{X}} q_0(\mathbf{y})g_{\mathbf{x}+(\hat{x}^i-x^i)\mathbf{e}_i-\mathbf{y}}(t)}{q_t(\mathbf{x})}$$

$$= \frac{\sum_{\mathbf{y} \in \mathcal{X}} q_0(\mathbf{y} + (\hat{x}^i - x^i)\mathbf{e}_i)g_{\mathbf{x}-\mathbf{y}}(t)}{q_t(\mathbf{x})}$$

$$= \sum_{\mathbf{y} \in \mathcal{X}} \frac{q_0(\mathbf{y})q_{t|0}(\mathbf{x}|\mathbf{y})}{q_t(\mathbf{x})} \cdot \frac{q_0(\mathbf{y} + (\hat{x}^i - x^i)\mathbf{e}_i)}{q_0(\mathbf{y})}$$

$$= \mathbb{E}_{\mathbf{y} \sim q_{0|t}(\cdot|\mathbf{x})} \frac{q_0(\mathbf{y} + (\hat{x}^i - x^i)\mathbf{e}_i)}{q_0(\mathbf{y})} \leq L,$$

where the last inequality is by Assumption 2 and $\mathbf{y} + (\hat{x}^i - x^i)\mathbf{e}_i$ should be understood in modulo $S$ sense. $\square$

### B.5 PROOF OF LEMMA 5

**Lemma 10.** *Suppose Assumption 2 holds. Let $\delta = 0$, then for all $i \in [d]$, $k \in \{0, 1, \cdots, K - 1\}$, $t \in [kh, (k+1)h]$, and $x \neq y \in [S]$, we have*

$$s_t^i(x)_y \leq \kappa_i, \quad and \quad \left| s_t^i(x)_y - s_{(k+1)h}^i(x)_y \right| \leq \kappa_i \cdot \frac{h}{1 - e^{-(k+1)h}}.$$

The proof of Lemma 10 is provided in Appendix C.

*Proof of Lemma 5.* Similar to the proof of Lemma 3, by utilizing the inequality $|a_1 a_2 - b_1 b_2| \leq |a_1 - b_1| a_2 + b_1 |a_2 - b_2| (a_1, a_2, b_1, b_2 \geq 0)$ for the first term in (24) and then combining Lemma 8, Lemma 9, and Lemma 10 with $\delta = 0$, we obtain

$$|s_t(\mathbf{x})_{i,\hat{x}^i} - s_{(k+1)h}(\mathbf{x})_{i,\hat{x}^i}| \leq \kappa_i \left[ \frac{1}{1 - e^{-(k+1)h}} h \cdot 2 + 8Sh \right] + 2h$$

$$\lesssim \left[ \frac{1}{1 - e^{-(k+1)h}} + S \right] \kappa_i h.$$

$\square$

## C OMITTED PROOFS IN APPENDIX B

### C.1 PROOF OF LEMMA 7

*Proof of Lemma 7.* Proposition 1 implies that

$$q_t^i = P_{0,t}^i \cdot p_{\text{data}}^i = e^{-t} p_{\text{data}}^i + (1 - e^{-t}) \pi.$$

Thus, we have

$$s_t^i(x)_y = \frac{q_t^i(y)}{q_t^i(x)} = \frac{p_{\text{data}}^i(y) + (e^t - 1)\frac{1}{S}}{p_{\text{data}}^i(x) + (e^t - 1)\frac{1}{S}}, \tag{25}$$

and

$$\left| s_t^i(x)_y - s_{(k+1)h+\delta}^i(x)_y \right| = \left| \frac{q_t^i(y)}{q_t^i(x)} - \frac{q_{(k+1)h+\delta}^i(y)}{q_{(k+1)h+\delta}^i(x)} \right|$$

$$= \frac{(e^{(k+1)h+\delta} - e^t)|p_{\text{data}}^i(x) - p_{\text{data}}^i(y)|}{(p_{\text{data}}^i(x) + \frac{1}{S}(e^{(k+1)h+\delta} - 1))(Sp_{\text{data}}^i(y) + (e^t - 1))}. \tag{26}$$

We can derive from (25) and (26) that if we retain the dependence on $\delta$, we obtain

$$s_t^i(x)_y \leq 1 + \frac{S}{e^t - 1} \leq 1 + \frac{S}{e^{kh+\delta} - 1},$$

and

$$\left| s_t^i(x)_y - s_{(k+1)h+\delta}^i(x)_y \right| \leq \frac{Se^{(k+1)h+\delta}(1 - e^{-((k+1)h+\delta-t)})}{(e^{(k+1)h+\delta} - 1)(e^t - 1)}$$

$$\leq \frac{Sh}{(1 - e^{-((k+1)h+\delta)})(e^{kh+\delta} - 1)}$$

$$\leq \frac{She^{-(kh+\delta)}}{(1 - e^{-(kh+\delta)})^2}.$$

$\square$

## C.2 PROOF OF LEMMA 8.

*Proof of Lemma 8.* First, we have

$$
|q^i_{t'|t}(x_{t'}|y) - q^i_{t'|(k+1)h+\delta}(x_{t'}|y)|
$$

$$
= \frac{1}{S}\left|(-1 + S \cdot \delta\{x_{t'}, y\})(e^{-(t'-t)} - e^{-(t'-((k+1)h+\delta))})\right|
$$

$$
\le e^{-(t'-((k+1)h+\delta))}(1 - e^{-((k+1)h+\delta-t)}) \le h,
$$

and

$$
q^i_{t'|t}(x_{t'}|x) = \frac{1}{S}(1 + e^{-(t'-t)}(-1 + S \cdot \delta\{x, x_{t'}\})) \ge \frac{1}{S}(1 - e^{-(t'-t)}) \ge \frac{1}{\sqrt{2}S};
$$

$$
q^i_{t'|t}(x_{t'}|x) \le \frac{1}{S}(1 + e^{-(t'-t)}(S - 1)) \le \frac{2}{S}
$$

for some large enough $t'$ (for example, $t' \ge T + 2\log\frac{S}{2}$). Hence, we have the estimate

$$
\left|\frac{q^i_{t'|t}(x_{t'}|y)}{q^i_{t'|t}(x_{t'}|x)} - \frac{q^i_{t'|(k+1)h+\delta}(x_{t'}|y)}{q^i_{t'|(k+1)h+\delta}(x_{t'}|x)}\right|
$$

$$
= \left|\frac{q^i_{t'|t}(x_{t'}|y)q^i_{t'|(k+1)h+\delta}(x_{t'}|x) - q^i_{t'|(k+1)h+\delta}(x_{t'}|y)q^i_{t'|t}(x_{t'}|x)}{q^i_{t'|t}(x_{t'}|x)q^i_{t'|(k+1)h+\delta}(x_{t'}|x)}\right| \le \frac{2h\frac{2}{S}}{\frac{1}{2S^2}} = 8Sh.
$$

For the other statement, it holds that

$$
\frac{q^i_{t'|t}(x_{t'}|y)}{q^i_{t'|t}(x_{t'}|x)} = \frac{1 + e^{-(t'-t)}(-1 + S \cdot \delta\{x_{t'}, y\})}{1 + e^{-(t'-t)}(-1 + S \cdot \delta\{x_{t'}, x\})} \le 1 + \frac{S}{e^{t'-t} - 1} \le 2
$$

for some large $t'$ (for example, $t' \ge T + \log S$). □

## C.3 PROOF OF LEMMA 9.

*Proof of Lemma 9.* By taking a large enough $t'$ (for example, $t' \ge T + d\log S + \log d$), we have

$$
2 \cdot D_{\text{TV}}(q_{t'|t}(\cdot|\mathbf{x}), q_{t'|(k+1)h+\delta}(\cdot|\mathbf{x})) = \sum_{\mathbf{x}_{t'}\in\mathcal{X}} |q_{t'|t}(\mathbf{x}_{t'}|\mathbf{x}) - q_{t'|(k+1)h+\delta}(\mathbf{x}_{t'}|\mathbf{x})|
$$

$$
= \sum_{\mathbf{x}_{t'}\in\mathcal{X}}\left|\prod_{i=1}^{d}q^i_{t'|t}(x^i_{t'}|x^i) - \prod_{i=1}^{d}q^i_{t'|(k+1)h+\delta}(x^i_{t'}|x^i)\right|
$$

$$
\le (S^d d e^{-(t'-((k+1)h+\delta))})h \le h,
$$

where the second last inequality comes from the inequality

$$
\left|\prod_{i=1}^{n}a_i - \prod_{i=1}^{n}b_i\right| \le \sum_{k=1}^{n}\left|(a_k - b_k)\prod_{i\ne k}\text{mix}\{a_i, b_i\}\right|
$$

where $\text{mix}\{a_i, b_i\}$ denotes taking a value from $a_i$ and $b_i$. □

## C.4 PROOF OF LEMMA 10

*Proof of Lemma 10.* We retain the dependence on $p^i_{\text{data}}$ in equations (25) and (26) with $\delta = 0$, and derive that

$$
s^i_t(x)_y \le \max\left\{\frac{(p^i_{\text{data}})_{\max}}{(p^i_{\text{data}})_{\min}}, 1\right\} = \kappa_i,
$$

and

$$
\left|s^i_t(x)_y - s^i_{(k+1)h}(x)_y\right| \le \frac{(p^i_{\text{data}})_{\max}}{(p^i_{\text{data}})_{\min}} \cdot \frac{h}{1 - e^{-(k+1)h}} = \kappa_i \cdot \frac{h}{1 - e^{-(k+1)h}}.
$$

□

---

**Algorithm 1** Generative Reverse Process Simulation through Uniformization

---

**Input:** Learned discrete score function $\hat{s}_{T-kh}$ ($k = 0, 1, \cdots, K-1$), total time $T$, discretization step $h > 0$, and $\delta = T - Kh \geq 0$

1: Draw $\mathbf{z}_0 \sim \pi^d$
2: **for** $k = 0, 1, \cdots, K-1$ **do**
3:     Set $\lambda_k = \max_{\mathbf{x} \in \mathcal{X}} \{\hat{s}_{T-kh}(\mathbf{x})_{\mathbf{x}}\}$
4:     Draw $M \sim \mathrm{Poisson}(\lambda_k h)$
5:     Set $\mathbf{y}_0 = \mathbf{z}_k$
6:     **for** $j = 0, 1, \cdots, M-1$ **do**
7:         Set $\mathbf{y}_{j+1} = \begin{cases} \mathbf{y}_j^{\backslash i} \odot \hat{y}^i, & \text{w.p. } \frac{\hat{s}_{T-kh}(\mathbf{y}_j)_{i,\hat{y}^i}}{\lambda_k}, \ 1 \leq i \leq d, \ \hat{y}^i \neq \mathbf{y}_j^i, \ \hat{y}^i \in [S] \\ \mathbf{y}_j, & \text{w.p. } 1 - \sum_{i=1}^d \sum_{\hat{y}^i \neq \mathbf{y}_j^i} \frac{\hat{s}_{T-kh}(\mathbf{y}_j)_{i,\hat{y}^i}}{\lambda_k} \end{cases}$
8:     **end for**
9:     Set $\mathbf{z}_{k+1} = \mathbf{y}_M$
10: **end for**
**Output:** A sample $\mathbf{z}_K$ from $p_{T-\delta}$

---

## D PRACTICAL ALGORITHM

Inspired by Chen & Ying (2024, Algorithm 1), we provide a practical generative sampling algorithm in Algorithm 1. The uniformization of CTMC guarantees the algorithm (Chen & Ying, 2024, Proposition 1). For convenience, we define $\hat{s}_t(\mathbf{x})_{\mathbf{x}} := \sum_{i=1}^d \sum_{\hat{x}^i \neq x^i} \hat{s}_t(\mathbf{x})_{i,\hat{x}^i}$ for $t \in [0, T]$, yielding that

$$\hat{Q}_t^{\leftarrow}(\mathbf{x}, \mathbf{x}) = -\sum_{\mathbf{y} \neq \mathbf{x}} \hat{Q}_t^{\leftarrow}(\mathbf{x}, \mathbf{y}) = -\sum_{i=1}^d \sum_{\hat{x}^i \neq x^i} \hat{Q}_t^{\leftarrow}(\mathbf{x}, \mathbf{x}^{\backslash i} \odot \hat{x}^i) = -\frac{1}{S} \sum_{i=1}^d \sum_{\hat{x}^i \neq x^i} s_{T-t}(\mathbf{x})_{i,\hat{x}^i}$$

$$= -\frac{1}{S} \hat{s}_{T-[\frac{t}{h}]h}(\mathbf{x})_{\mathbf{x}}.$$

Note that we run a Poisson point process to sample based on the transition probability matrix $\exp\left(h\hat{Q}_{hk}^{\leftarrow}\right)$ in each iteration. Running Algorithm 1 requires $M \sim \mathrm{Poisson}(\lambda)$ steps with $\lambda = \sum_{k=0}^{K-1} \lambda_k h = (\sum_{k=0}^{K-1} \max_{\mathbf{x} \in \mathcal{X}} \{\hat{s}_{T-kh}(\mathbf{x})_{\mathbf{x}}\})h$, which characterizes the sampling complexity.

**Discussion on $\lambda_k$ in Algorithm 1.** We set $\lambda_k = \max_{\mathbf{x} \in \mathcal{X}} \{\hat{s}_{T-kh}(\mathbf{x})_{\mathbf{x}}\}$ in Algorithm 1, where the maximum is taken over the discrete set $\mathcal{X} = [S]^d$. When $|\mathcal{X}| = S^d$ is so large that it is impractical to obtain this exact maximum, by the uniformization of CTMC (Chen & Ying, 2024, Proposition 1), we can instead set $\lambda_k$ as an upper bound for $\max_{\mathbf{x} \in \mathcal{X}} \{\hat{s}_{T-kh}(\mathbf{x})_{\mathbf{x}}\}$. Since we know that $\sum_{i=1}^d \sum_{\hat{x}^i \neq x^i} s_{T-kh}(\mathbf{x})_{i,\hat{x}^i} \leq dS(1 + \frac{S}{e^{T-kh}-1})$ for all $k \in \{0, 1, \cdots, K-1\}$ and $\mathbf{x} \in \mathcal{X}$ by Lemma 2, we can apply score clipping to ensure that $\hat{s}_{T-kh}(\mathbf{x})_{\mathbf{x}} \leq \frac{3}{2} dS(1 + \frac{S}{e^{T-kh}-1})$ for all $\mathbf{x} \in \mathcal{X}$. Therefore, we can set $\lambda_k = \frac{3}{2} dS \left(1 + \frac{S}{e^{T-kh}-1}\right)$ which is tractable given any $k, T, h, d$, and $S$, thereby avoiding the need to calculate the maximum over the set $\mathcal{X}$.

**Discussion on $\lambda$.** By applying score clipping with $\hat{s}_{T-kh}(\mathbf{x})_{\mathbf{x}} \leq \frac{3}{2} dS(1 + \frac{S}{e^{T-kh}-1})$, we have

$$\lambda \lesssim \sum_{k=0}^{K-1} dS \left(1 + \frac{S}{e^{T-kh}-1}\right) h = dST + dS^2 h \sum_{k=1}^K \frac{1}{e^{kh+\delta}-1}$$

$$\lesssim dST + dS^2 h \int_{h+\delta}^T \frac{1}{e^x - 1}\, \mathrm{d}t \lesssim dS(T + Sh\log(1/(h+\delta))).$$

By choosing $T$ and $h$ in Corollary 1, we have

$$\lambda \lesssim dS \left(\log \frac{d\log S}{\epsilon} + \min\left\{\delta\left(\frac{\epsilon}{C_1 S^2 d}\right)^{\frac{1}{3}}, \left(\frac{\epsilon}{C_1 S^2 d}\right)^{\frac{1}{2}}\right\} S\log(1/\delta)\right)$$

$$\leq dS \log \frac{d \log S}{\epsilon} + \left( \frac{\epsilon}{C_1} \right)^{\frac{5}{12}} S^{\frac{7}{6}} d^{\frac{7}{12}} \delta^{\frac{1}{2}} \log(1/\delta), \tag{27}$$

where we used the inequality $\min\{a, b\} \leq \sqrt{ab}$ for $a, b \geq 0$; by choosing $T$ and $h$ in Corollary 2, we have

$$\lambda \lesssim dS \left( \log \frac{d \log S}{\epsilon} + Sh \log(1/h) \right) \lesssim dS \left( \log \frac{d \log S}{\epsilon} + \sqrt{\frac{\epsilon}{C_2 \kappa^2}} \log \frac{C_2 S^2 \kappa^2}{\epsilon} \right), \tag{28}$$

which depends on the property of data distribution. When score clipping is applied as discussed in Appendix A.4, we can specify $C_1 = \frac{3S}{\delta}$ and $C_2 = \frac{3}{2} L$ and further obtain from (27) that

$$\lambda \lesssim dS \log \frac{d \log S}{\epsilon} + \epsilon^{\frac{5}{12}} S^{\frac{19}{12}} d^{\frac{7}{12}} \delta^{\frac{1}{12}} \log(1/\delta) \to dS \log \frac{d \log S}{\epsilon} \quad (\delta \to 0^+) \tag{29}$$

with early stopping, and from (28) that

$$\lambda \lesssim dS \left( \log \frac{d \log S}{\epsilon} + \sqrt{\frac{\epsilon}{L \kappa^2}} \log \frac{L \kappa^2 S^2}{\epsilon} \right) \lesssim dS \log \frac{d \log S}{\epsilon} + 1 + \sqrt{\frac{\epsilon}{L \kappa^2}} \log S \tag{30}$$

without early stopping. Note that the last term of (30) can be rather small for some large $L$ and $\kappa^2$, leading the first term to be the dominant term in the bound (30), which matches the bound (29) for the sampling complexity with a sufficient small $\delta > 0$.

## E  BREGMAN DIVERGENCE

**Definition 1** (Bregman divergence). *Let $\phi$ be a strictly convex function defined on a convex set $\mathcal{S} \subset \mathbb{R}^n$ ($n \in \mathbb{N}_+$) and $\phi$ is differentiable. The Bregman divergence $D_\phi(x \| y) : \mathcal{S} \times \mathcal{S} \to \mathbb{R}_+$ is defined as*

$$D_\phi(x \| y) = \phi(x) - \phi(y) - \nabla \phi(y)^\top (x - y).$$

In particular, the generalized I-divergence

$$D_I(\mathbf{x} \| \mathbf{y}) = \sum_{i=1}^n \left[ -x^i + y^i + x^i \log \frac{x^i}{y^i} \right]$$

is generated by the negative entropy function $I(\mathbf{x}) = \sum_{i=1}^n x^i \log x^i$. When restricted on the simplex, the generalized I-divergence becomes KL divergence.

The Bregman divergence does not satisfy the triangle inequality. However, for the negative entropy restricted to a closed box contained in $\mathbb{R}_+^n$, we have the following proposition, which provides an analogous form of triangle inequality.

**Proposition 3.** *Let the negative entropy function $I(\mathbf{x}) = \sum_{i=1}^n x^i \log x^i$ defined on $[\frac{1}{C}, C]^n$ ($C > 0$). Then for all $\mathbf{x}, \mathbf{y}, \mathbf{z} \in [\frac{1}{C}, C]^n$, we have*

$$D_I(\mathbf{x} \| \mathbf{y}) \leq C \cdot \|\mathbf{x} - \mathbf{z}\|_2^2 + 2C^2 \cdot D_I(\mathbf{z} \| \mathbf{y}).$$

*Proof of Proposition 3.* By $\nabla^2 I(\mathbf{x}) = \text{diag}\{\frac{1}{x^1}, \cdots, \frac{1}{x^n}\} \preceq C I_n$ for all $\mathbf{x} \in [\frac{1}{C}, C]^n$, there exists some $\theta \in [0, 1]$ such that for all $\mathbf{x}, \mathbf{y}, \mathbf{z} \in [\frac{1}{C}, C]^n$, it holds that

$$\begin{aligned}
D_I(\mathbf{x} \| \mathbf{y}) &= \frac{1}{2} (\mathbf{x} - \mathbf{y})^\top \nabla^2 I(\mathbf{y} + \theta(\mathbf{x} - \mathbf{y}))(\mathbf{x} - \mathbf{y}) \\
&\leq \frac{C}{2} \|\mathbf{x} - \mathbf{y}\|_2^2 \\
&\leq C \cdot (\|\mathbf{x} - \mathbf{z}\|_2^2 + \|\mathbf{z} - \mathbf{y}\|_2^2) \\
&\leq C \cdot \|\mathbf{x} - \mathbf{z}\|_2^2 + 2C^2 \cdot D_I(\mathbf{z} \| \mathbf{y}),
\end{aligned}$$

where the last inequality is by the strongly-convexity of $I$ on $[\frac{1}{C}, C]^n$ since $\nabla^2 I(\mathbf{x}) = \text{diag}\{\frac{1}{x^1}, \cdots, \frac{1}{x^n}\} \succeq \frac{1}{C} I_n$. $\qquad \square$

