# OpenReview forum: "Convergence of Score-Based Discrete Diffusion Models: A Discrete-Time Analysis"
_ICLR.cc/2025/Conference — ICLR 2025 Poster_

### Official Review · Reviewer_k765 · 2024-10-31

**Soundness:** 4
**Presentation:** 4
**Contribution:** 4
**Rating:** 8
**Confidence:** 3

**Summary:**

This work develops an algorithm for score-based diffusion models over a discrete state space. Their new algorithm relies on the uniformization of the CTMC developed by Chen and Ying and discretization in time. The main innovation of the work is theoretical. Using this new algorithm, the authors can derive bounds on the generated samples under less restrictive assumptions than in previous work. The main assumption is an approximate score function, where analogous to the results on continuous denoising diffusion models, the authors are able to develop a bound with early stopping. Under the further assumption of bounded scores for the data distribution, the authors show that early stopping is no longer needed and can derive bounds at $t=0$.

**Strengths:**

The paper is well-written and represents a strong contribution to the generative model literature. The assumptions are weaker than those for existing algorithms. There is a strong case for accepting this paper due to its novel theoretical analysis of an important problem. The theoretical framework given may be useful in analyzing more general settings and motivating further algorithmic improvements.

**Weaknesses:**

First, It is unclear how tight these bounds are. Can the authors compare these to bounds in the continuous setting? Second, I don’t understand everything in the table, and it would be better to flesh out the comparison with Chen and Ying. In particular, their assumptions are different, can you tell us what they are? Finally, there are no tests of the algorithm in practice. While the theoretical contribution is nice, it remains to be seen if it will have an impact on practice.

**Questions:**

Can the authors provide more explanation comparing their result to the Chen and Ying result?

Also, can they provide a comparison between their guarantees and the guarantees for continuous models? I am curious if they are analogous or what the fundamental differences are.

---

> ### Author Response · Authors · 2024-11-20
> **Part 1**
>
> Thank you for the strong support and valuable suggestion!
>
> ---
>
> **Q1**. Can the authors provide more explanation comparing their result to the Chen and Ying result?
>
> **A1**. We present a detailed comparison between our results and Chen and Ying (2024) as follows (see Table 1 in our paper):
> 1. Time-discretized: By utilizing the uniformization technique of CTMC, Chen and Ying (2024) proposed a sampling algorithm that exactly simulates the reverse CTMC. Hence, their convergence bounds get rid of the discretization error term. However, our algorithm discretizes the time and utilizes score estimators on the pre-defined time steps, which introduces a discretization error in the convergence bound that scales with the step size.
>
> 2. Score error assumption: Chen and Ying (2024) made the score error assumption: $\int_\delta^T \mathbb{E}\_{x_t\sim q_t} D_I(s_t(x_t) \Vert \hat{s}\_t(x_t)) \mathrm{d}t \leq \epsilon$ . Since this assumption is made over the time interval $[\delta, T]$ with continuous integral on the time which contains the true score functions and the score estimators across the time interval $[\delta, T]$, we, therefore, refer to this assumption as the "continuous score error" in Table 1. In contrast, since we discretize the time in our sampling algorithm in our paper, we make the following score error assumption: $\frac{1}{S} \sum_{k=0}^{K-1} \int_{kh+\delta}^{(k+1)h+\delta} \mathbb{E}_{x_t\sim q_t} D_I(s\_{(k+1)h+\delta}(x_t) \Vert \hat{s}\_{(k+1)h+\delta}(x_t)) \mathrm{d}t \leq \epsilon$ , which contains the true score functions and the score estimators at the predefined time steps. We refer to this assumption as the "discretized score error" in Table 1, which can be viewed as a time-discretization of score entropy.
>
> 3. Chen and Ying (2024) assumed the bounded score estimator to implement their sampling algorithm. Though we did not make this assumption in the maintext, for the practical implementation of our Algorithm 1, the application of the score clipping discussed in the Appendix essentially applies this bounded score estimator assumption. Namely, since we have known the true bound of the score function, we can ensure that the score estimators are also bounded by score clipping.
>
> 4. Both Chen and Ying (2024) and our paper derived the results with and without early stopping.  Our algorithm discretizes the time to simulate the reverse process, introducing an additional discretization error term in our bounds. Furthermore, our convergence analysis is conducted in a more general $[S]^d$ setting, as opposed to the $\\{0,1\\}^d$ hypercube framework used in their work.
>
> ---
>
> **Q2**. Also, can they provide a comparison between their guarantees and the guarantees for continuous models? I am curious if they are analogous or what the fundamental differences are.
>
> **A2**. We provide a comparison between our convergence results for discrete diffusion models and the continuous diffusion models as follows. We choose two typical convergence bounds achieved for continuous diffusion models.
>
> Continuous state:
>   1. Chen et al. (2023b): $D\_\mathrm{KL}(q_0\Vert p_T)\lesssim e^{-2T}D\_\mathrm{KL}(q_0\Vert p\_\mathrm{data})+(\epsilon+L^2 dh+L^2 m_2^2 h^2)T$, where $L$ is the Lipschitz constant of the score function, $m_2$ is the second moment bound of the data distribution, and $h$ is the step size.
>
>   2. Benton et al. (2024a): $D\_\mathrm{KL}(q_\delta\Vert p\_{T-\delta})\lesssim d e^{-2T}+\kappa^2 dN+\kappa dT+\epsilon$, where $\kappa$ is related to the time steps, and $N$ is the number of time steps.
>
> Discrete state:
>   1. Chen and Ying (2024): $D\_\mathrm{KL}(q_\delta\Vert p\_{T-\delta})\lesssim de^{-T}+\epsilon $
>   2. Ours: $D\_\mathrm{KL}(q_\delta\Vert p\_{T-\delta})\lesssim de^{-T}\log S+\delta^{-3}C_1 S^2h^3 d+C_1 S^2 h^2 dT+C_1^2\epsilon$ (with early stopping)
>                 $D_\mathrm{KL}(q_0\Vert p_T)\lesssim de^{-T}\log S+C_2 S^2 h^2 \kappa^2 T+C_2^2\epsilon$ (without early stopping)
>
> These convergence bounds are analogous, for the main KL divergence bounds for convergence comprise three components: score estimation error, discretization error, and the error due to insufficient mixing of the forward process. This is because the exponential integrator sampling procedure in continuous diffusion models discretized the true reverse SDE, and our sampling algorithms discretized the true reverse CTMC, both leading to a discretization error term. We remark that the appearance of the quantities $C_1$ and $C_2$ in our bound, which is absent in the continuous SDE counterpart, stems from the lack of triangle inequality for the Bregman divergence versus $L^2$ distance in the score-matching objective. Hence, we note that a significant difference between the discrete CMTC framework and the continuous SDE framework is the score-matching methods. In addition, we note that our main bounds are nearly linear in the dimension $d$, matching the best result found in continuous diffusion models (Benton et al., 2024a).

---

> ### Author Response · Authors · 2024-11-20
> **Part 2**
>
> **Q3.** There are no tests of the algorithm in practice. While the theoretical contribution is nice, it remains to be seen if it will have an impact on practice
>
> **A3.**  We appreciate your strong support of our theoretical contribution. While our focus is theoretical, we build upon the CTMC framework that has been empirically validated to be effective in previous works (Campbell et al., 2022; Lou et al., 2024). Our paper belongs to the theoretical line of research in diffusion models, similar to the analysis of continuous diffusion models (Chen et al., 2023b; Benton et al., 2024a) and discrete diffusion models (Chen & Ying, 2024)
>
> ---
> **References**
>
> [1] Sitan Chen, Sinho Chewi, Jerry Li, Yuanzhi Li, Adil Salim, and Anru Zhang. Sampling is as easy as learning the score: theory for diffusion models with minimal data assumptions. In The Eleventh  International Conference on Learning Representations, 2023b.
>
> [2] Joe Benton, Valentin De Bortoli, Arnaud Doucet, and George Deligiannidis. Nearly d-linear convergence bounds for diffusion models via stochastic localization. In The Twelfth International  Conference on Learning Representations, 2024a.
>
> [3] Hongrui Chen and Lexing Ying. Convergence analysis of discrete diffusion model: Exact implementation through uniformization. arXiv preprint arXiv:2402.08095, 2024.
>
> [4] Campbell, Joe Benton, Valentin De Bortoli, Thomas Rainforth, George Deligiannidis, and Arnaud Doucet. A continuous time framework for discrete denoising models. Advances in Neural Information Processing Systems, 35:28266–28279, 2022.
>
> [5]  Aaron Lou, Chenlin Meng, and Stefano Ermon. Discrete diffusion language modeling by estimating the ratios of the data distribution. In International Conference on Machine Learning, 2024.

---

> > ### Comment · Reviewer_k765 · 2024-11-26
> >
> > I appreciate the authors detailed response to my questions, and I maintain that the paper is worthy of acceptance to ICLR.

---

> > > ### Author Response · Authors · 2024-11-26
> > > **Thank you**
> > >
> > > Thank you for your support. We're glad that our rebuttal has addressed your questions. Your thoughtful comments and feedback have been invaluable to us.

---

### Official Review · Reviewer_nnaP · 2024-11-02

**Soundness:** 3
**Presentation:** 3
**Contribution:** 3
**Rating:** 6
**Confidence:** 5

**Summary:**

This paper provides the convergence rate of the discrete diffusion models introduced by Lou and Ermon. The authors consider the finite state space $S^d$, extending early analysis by Chen and Ying in the continuous time.

**Strengths:**

The paper provides a timely convergence analysis of the discrete diffusion models, which is important in understanding this new class of models (with applications to LMM). I checked most proofs, and they are scientifically correct.

**Weaknesses:**

The weaknesses are:

(1) The main idea of proofs are very similar to Chen and Yin, with the key to the proof the representation given by Proposition 1. The differences are: (1) the paper extends the previous results from $\{0,1\}^d$ to $S^d$; (2) the paper considers the issue of discretization. I agree that (2) is important, while (1) seems to me incremental.

(2) The authors made Assumption 2, which requires the ratio constant $L$ is independent of $d$. I am dubious on this assumption, since the main application of the discrete diffusion models is on the LLM. I would encourage the authors to provide a numerical analysis to show indeed this assumption is valid.

(3) Regarding early stopping: Chen and Yin considers early stopping since they are concerned with the continuous dynamics. It is known that the score matching has large errors near time $0$, and even the continuous dynamics may not be well-defined at $0$. Since the paper studies the discrete scheme, I wonder if there is a particular reason why the authors also consider early stopping.

(4) Except the error for the initialization, which uses log-Sobolev inequality (more or less expected), the other ingredients in the proof are very similar to the existing approaches to the SDE-based diffusion models. Of course, the authors dealt with CDMC, which is slightly different.

**Questions:**

Please see the weakness section.

---

> ### Author Response · Authors · 2024-11-20
> **Part 1**
>
> We appreciate the reviewer's support and valuable comments.
>
> ---
>
> **Q1**. The main idea of proofs are very similar to Chen and Ying, with the key to the proof the representation given by Proposition 1. The differences are: (1) the paper extends the previous results from $\\{0,1\\}^d$ to $[S]^d$; (2) the paper considers the issue of discretization. I agree that (2) is important, while (1) seems to me incremental.
>
> **A1**. As detailed in Section 6, our proof techniques include three key components: (A) establishing exponential convergence of the forward process with uniform mixing in the general state space $[S]^d$, (B) employing a Girsanov-based method to analyze the discretization error, and (C) deriving novel score movement bounds to control the time-variation of score functions.
>
> Regarding point (1): In practice, the discrete state generally factorizes into $[S]^d$, which fits many generative modeling tasks including text generation, image generation, and protein design. This makes our state space setting and sampling algorithm more broadly applicable, as opposed to the $\\{0,1\\}^d$ hypercube framework used by Chen and Ying (2024).
>
> Regarding point (2): The analysis of discretization error in our discrete-time sampling algorithm is both the most challenging aspect and an important contribution of our work. In particular, our proof leverages Girsanov's Theorem to bound the path measure KL divergence between the true reverse process and the sampling process, while deriving score movement bounds represents a key and novel component of our convergence analysis.
>
> ---
>
> **Q2**. The authors made Assumption 2, which requires the ratio constant L is independent of d. I am dubious on this assumption, since the main application of the discrete diffusion models is on the LLM. I would encourage the authors to provide a numerical analysis to show indeed this assumption is valid.
>
> **A2**. We appreciate the reviewer's insights. For the validation of Assumption 2, we provide the following explanation. Assumption 2 is satisfied in many cases, e.g., when the data distribution $p\_\mathrm{data}$ is the product of independent and identically distributed (i.i.d.) components, each with a marginal distribution $p\_\mathrm{data}^i (i\in [d])$ fully supported on $[S]$. In this case, the uniform upper bound $L$ in Assumption 2 is thus the uniform upper bound of the score function for $p\_\mathrm{data}^i$. Importantly, $L$ depends only on $S$ and remains independent of $d$. We have added this supplementary explanation for Assumption 2 to the paper in Lines 378-379 (highlighted in blue).
>
> In addition, we would like to note that Campbell et al. (2022) also made a similar assumption that requires the score uniform bound $L$ is dependent on $S$ but not on $d$. As they pointed out, the uniform boundness for the data distribution follows trivially from the strict positiveness of $p\_\mathrm{data}$ if we allow $L$ to depend on $d$, and it also makes explicit the dependence of the error bound on the dimension $d$. We adopt this assumption which enables us to derive a bound that can be nearly linear in $d$ without early stopping, as stated in Theorem 2.
>
> ---
>
> **Q3**. Regarding early stopping: Chen and Yin considers early stopping since they are concerned with the continuous dynamics. It is known that the score matching has large errors near time 0, and even the continuous dynamics may not be well-defined at 0. Since the paper studies the discrete scheme, I wonder if there is a particular reason why the authors also consider early stopping.
>
> **A3**. Indeed, the large errors in score matching near time $t=0$ present a significant challenge. As explained in Lines 299 and 371–372 of our paper, the discrete score function can diverge as $t\to 0$ for data distributions lacking full support on $[S]^d$. The unbounded score at $t=0$ prevents us from establishing uniform upper bounds for the score and its estimators, which are critical for the validity of our key Lemmas 2, 3, 4, and 5. To address this issue, we avoid the zero time point by performing early stopping. However, if we assume that the data distribution has full support on $[S]^d$, the early stopping can be removed and we can derive a convergence bound without early stopping.

---

> > ### Author Response · Authors · 2024-11-20
> > **Part 2**
> >
> > **Q4**.  Except the error for the initialization, which uses log-Sobolev inequality (more or less expected), the other ingredients in the proof are very similar to the existing approaches to the SDE-based diffusion models. Of course, the authors dealt with CDMC, which is slightly different.
> >
> > **A4**. The discretization error terms in our main bounds are novel in the literature on score-based discrete diffusion models. Specifically, in the CTMC setting, we are the first to establish a score movement bound for high-dimensional discrete score functions. This result forms a critical component of our analysis of the discretization error.
> >
> > In addition, our discrete-time sampling algorithm and methodology are closely related to the previous works on the theory of continuous diffusion models within the SDE framework (Chen et al., 2023b; Benton et al., 2024b), as pointed out in Lines 74-76 of our paper. Inspired by the exponential integrator approach in the SDE framework, our sampling algorithm adopts a similar procedure, and the proof of its convergence leverages techniques analogous to those used for SDE-based diffusion models including Girsanov's Theorem and theoretical properties of score functions such as score movement bound.
> >
> > ---
> >
> > **References**
> >
> > [1] Hongrui Chen and Lexing Ying. Convergence analysis of discrete diffusion model: Exact implementation through uniformization. arXiv preprint arXiv:2402.08095, 2024.
> >
> > [2] Sitan Chen, Sinho Chewi, Jerry Li, Yuanzhi Li, Adil Salim, and Anru Zhang. Sampling is as easy as  learning the score: theory for diffusion models with minimal data assumptions. In The Eleventh  International Conference on Learning Representations, 2023b.
> >
> > [3] Joe Benton, Yuyang Shi, Valentin De Bortoli, George Deligiannidis, and Arnaud Doucet. From  denoising diffusions to denoising markov models. Journal of the Royal Statistical Society Series  B: Statistical Methodology, 86(2):286–301, 2024b.

---

### Official Review · Reviewer_qT9k · 2024-11-04

**Soundness:** 4
**Presentation:** 4
**Contribution:** 3
**Rating:** 8
**Confidence:** 3

**Summary:**

The paper  examines theoretical aspects of score-based discrete diffusion models within a Continuous Time Markov Chain (CTMC) framework. The authors aim to address the underexplored convergence properties of these models, especially in discrete state spaces, compared to their continuous counterparts which have been widely studied. Key contributions of the paper include: 1. The authors propose a discrete-time sampling algorithm designed for high-dimensional discrete diffusion tasks. This algorithm leverages score estimators at specific time points to approximate the reverse process of the diffusion model. 2. They provide convergence bounds for the KL divergence and TV distance between the generated sample distribution and the target data distribution. The bounds are derived for cases both with and without early stopping, depending on assumptions about the data distribution's properties. 3.The convergence analysis is performed using a Girsanov-based method, which enables the authors to assess the score estimation error, discretization error, and mixing properties of the forward process. This method is adapted from techniques in continuous diffusion models and tailored to the discrete setting. 4. The paper establishes that their convergence bounds scale nearly linearly with the data dimension, aligning with the best results for continuous models. Additionally, they discuss practical considerations, such as the need for score clipping and early stopping under certain conditions to handle potential score function divergences. 5. The authors compare their method with prior approaches, particularly highlighting differences in sampling efficiency and the elimination of certain assumptions on score estimators, thus broadening the algorithm's applicability in discrete settings.

**Strengths:**

Overrall I think this work is clear and studies a clean problem which is relevant to the theoretical understanding of diffusion models.. Theorems 1 and 2 derive rigorous convergence bounds for score-based discrete diffusion models using KL divergence (Theorem 1 with early stopping, Theorem 2 without early stopping). This provides a critical theoretical foundation for discrete diffusion modeling, aligning nearly linearly with the dimension $d$, a promising result compared to continuous models. And the use of a Girsanov-based approach to analyze the sampling algorithm in a discrete setting is particularly noteworthy (Sections 5 and 6). This method adapts well-established techniques from continuous diffusion models and is a creative application in the discrete domain, enabling a novel convergence analysis.
The work also proposes a time-discretized approach (Section 4 and Algorithm 1), which involves sampling at discrete time steps rather than simulating the continuous CTMC path. This approach allows for more efficient sampling, as it does not require continuous access to the reverse CTMC. Additionally, to ensure bounded score estimates, the authors introduce score clipping as a practical solution for handling extreme values (discussed in Section 5).

**Weaknesses:**

I don't see major weaknesses of this work, but I do have some comments:
1. Since the authors claim improved sampling efficiency (e.g., fewer function evaluations), would compare actual runtime or convergence speed with other discrete sampling methods (such as the uniformization technique in Chen & Ying) substantiate these claims?
2. Assumption 2 requires the data distribution to have full support and uniform bounds. This may restrict the model.
3. Although the work mentions similarities with continuous diffusion models, there is limited discussion on when to use their discrete CTMC approach versus discretized continuous models for discrete data.
4. Generalizability to Other State Spaces: The works' algorithm assumes a general state space $[S]^d$, but would it be insightful to discuss how it might extend to more complex or structured discrete spaces, such as graph-based or hierarchical state spaces, which are common in discrete data applications?

**Questions:**

1. Have the authors considered adaptive step sizes as a way to minimize discretization error? If so, could author's share insights on how this might affect the overall convergence bound?
2. The paper proposes a CTMC-based discrete diffusion model, but certain continuous diffusion approaches are also adaptable to discrete data. Could the authors elaborate on specific scenarios or types of tasks where their CTMC-based model would be preferred over these alternative approaches?
3. Also see weaknesses part above

---

> ### Author Response · Authors · 2024-11-20
> **Part 1**
>
> Thank you for your strong support and valuable feedback!
>
> ---
>
> **Q1**. Since the authors claim improved sampling efficiency (e.g., fewer function evaluations), would compare actual runtime or convergence speed with other discrete sampling methods (such as the uniformization technique in Chen & Ying) substantiate these claims?
>
> **A1**.  We appreciate the reviewer's comment and acknowledge that our claim about improved efficiency requires revision. Although we derived an approximate sampling complexity of $\text{Poisson}(\lambda)\  \text{where}\  \lambda =O(dS\log\frac{d\log S}{\epsilon})$ for our algorithm, i.e., running our sampling algorithm requires $M\sim \text{Poisson}(\lambda)$ steps through uniformization (see Discussion on $\lambda$ in Appendix D in our paper), this does not quantitatively demonstrate that our algorithm is more efficient than other discrete sampling methods, such as the one in Chen and Ying (2024), which has a sampling complexity of $\text{Poisson}(\lambda)$ with $\lambda=O(d\log\frac{d}{\epsilon^{4/3}})$. We have highlighted the following revision in blue on Lines 423-424: our algorithm discretizes the time to simulate the reverse process and calls the score estimator at fixed discretization points $\\{kh+\delta\\}\_{k\in [K]}$ instead of randomly sampled times, as done in Chen and Ying (2024).
>
> ---
>
> **Q2**. Assumption 2 requires the data distribution to have full support and uniform bounds. This may restrict the model.
>
> **A2**.  Assumption 2 is an assumption for the data distribution to give a further convergence result that is nearly linear in the dimension $d$ without early stopping. However, Assumption 2 holds in many cases, e.g., if the data distribution is the product of independent and identically distributed (i.i.d.) components, each with a marginal distribution fully supported on $[S]$. We have added this supplementary explanation for Assumption 2 in our paper on Lines 378-379 (highlighted in blue). Additionally, we note that similar assumptions are also made by Chen and Ying (2024) (aimed to remove early stopping) and Campbell et al. (2022) for theoretical analysis.
>
> ---
>
> **Q3**. Generalizability to Other State Spaces: The works' algorithm assumes a general state space $[S]^d$, but would it be insightful to discuss how it might extend to more complex or structured discrete spaces, such as graph-based or hierarchical state spaces, which are common in discrete data applications?
>
> **A3**. We appreciate the reviewer’s insightful question. Extending our methodology and analysis to more complex or structured discrete spaces is promising and interesting. In each specific setting, exploring the corresponding discrete score function and score-matching methods would be essential to design sampling algorithms. For graph-based state spaces, our framework shows direct applicability. Building on Vignac et al. (2022), we can perform the diffusion processes separately on each node and edge feature within a graph setting. In this setting, the state space comprises the discrete node types $\mathcal{X}$ and edge types $\mathcal{E}$, which can be addressed as instances of our one-dimensional state space framework where $d=1$. For any given node or edge, we can define distinct CTMC diffusion processes and reverse sampling processes. Investigating and validating this extension would be an exciting direction for future research.
>
> ---
>
> **Q4**. Have the authors considered adaptive step sizes as a way to minimize discretization error? If so, could author's share insights on how this might affect the overall convergence bound?
>
> **A4**. As an initial exploration of discrete-time sampling algorithms and corresponding convergence analyses for score-based discrete diffusion models, we did not incorporate adaptive step sizes, as proposed by Chen et al. (2023a) and Benton et al. (2024a) for continuous diffusion models. Instead, we followed earlier works on the convergence of continuous diffusion models and adopted a constant step size for analytical simplicity, as done in Chen et al. (2023b) and Yang and Wibisono (2022).  Scheduling the step sizes indeed has the potential to decrease the discretization error. According to Lemma 3 in our paper, when $k$ is small, the score movement bound can be very large, particularly for unbalanced data distributions, leading to a significant discretization error term when $t$ is small. Hence, analogous to the exponential decay step size schedules suggested by Chen et al. (2023a) and Benton et al. (2024a), applying a decaying step size schedule to slow down the CTMC as $t\to 0$ may help decrease the overall discretization error. The critical challenge in determining an optimal step size schedule lies in deriving the score movement bound for general step sizes, which may involve a complex mathematical formulation and requires further investigation. We would leave this as a direction for future research.

---

> > ### Author Response · Authors · 2024-11-20
> > **Part 2**
> >
> > ---
> >
> > **Q5**. The paper proposes a CTMC-based discrete diffusion model, but certain continuous diffusion approaches are also adaptable to discrete data. Could the authors elaborate on specific scenarios or types of tasks where their CTMC-based model would be preferred over these alternative approaches?
> >
> > **A5**. Text generation is a typical task where CTMC-based discrete diffusion models are particularly well-suited due to the inherently discrete nature of the text. Zheng et al. (2023) empirically showed that the Discrete Denoising Diffusion Probabilistic Model (D3PM) (Austin et al., 2021) outperforms continuous diffusion models for text generation tasks (see Table 1 in their paper). Furthermore, Campbell et al. (2022) demonstrated that the CTMC framework offers significantly greater flexibility in defining reverse sampling schemes compared to discrete-time diffusion model approaches such as D3PM. This implies the advantages of using the CTMC framework over continuous diffusion methods for text generation. Additionally, Lou et al. (2024) showed that CTMC-based discrete diffusion models are competitive with autoregressive models, achieving superior performance over GPT-2 in particular. These empirical studies suggest that CTMC-based models would be preferred over these alternative approaches for discrete data generation tasks.
> >
> > ---
> >
> > **References**
> >
> > [1] Jacob Austin, Daniel D Johnson, Jonathan Ho, Daniel Tarlow, and Rianne Van Den Berg. Structured denoising diffusion models in discrete state-spaces. Advances in Neural Information Processing  Systems, 34:17981–17993, 2021.
> >
> > [2] Joe Benton, Valentin De Bortoli, Arnaud Doucet, and George Deligiannidis. Nearly d-linear convergence bounds for diffusion models via stochastic localization. In The Twelfth International Conference on Learning Representations, 2024a.
> >
> > [3] Andrew Campbell, Joe Benton, Valentin De Bortoli, Thomas Rainforth, George Deligiannidis, and  Arnaud Doucet. A continuous time framework for discrete denoising models. Advances in Neural  Information Processing Systems, 35:28266–28279, 2022.
> >
> > [4] Lin Zheng, Jianbo Yuan, Lei Yu, and Lingpeng Kong. A reparameterized discrete diffusion model for text generation. arXiv preprint arXiv:2302.05737,2023.
> >
> > [5] Clement Vignac, Igor Krawczuk, Antoine Siraudin, Bohan Wang, Volkan Cevher, and Pascal Frossard. Digress: Discrete denoising diffusion for graph generation. arXiv preprint arXiv:2209.14734, 2022
> >
> > [6] Aaron Lou, Chenlin Meng, and Stefano Ermon. Discrete diffusion language modeling by estimating the ratios of the data distribution. In International Conference on Machine Learning, 2024.
> >
> > [7] Hongrui Chen and Lexing Ying. Convergence analysis of discrete diffusion model: Exact implementation through uniformization. arXiv preprint arXiv:2402.08095, 2024.
> >
> > [8] Hongrui Chen, Holden Lee, and Jianfeng Lu. Improved analysis of score-based generative modeling:  User-friendly bounds under minimal smoothness assumptions. In International Conference on Machine Learning, pages 4735–4763. PMLR, 2023a.
> >
> > [9] Sitan Chen, Sinho Chewi, Jerry Li, Yuanzhi Li, Adil Salim, and Anru Zhang. Sampling is as easy as learning the score: theory for diffusion models with minimal data assumptions. In The Eleventh International Conference on Learning Representations, 2023b.
> >
> > [10] Kaylee Yingxi Yang and Andre Wibisono. Convergence of the inexact langevin algorithm and score-based generative models in kl divergence. arXiv preprint arXiv:2211.01512,2022.

---

### Official Review · Reviewer_Gwu3 · 2024-11-04

**Soundness:** 3
**Presentation:** 2
**Contribution:** 3
**Rating:** 6
**Confidence:** 4

**Summary:**

The paper presents a discrete-time sampling method using score estimators at fixed points in a multidimensional state space. It provides bounds on how closely the sample matches the data distribution, with KL divergence bounds nearly linear in dimension, comparable to top diffusion models. The analysis uses a Girsanov-based approach to define key properties of the discrete score function for effective sampling.

**Strengths:**

The paper tackles an interesting theoretical problem. Theoretical analysis of diffusion models with finite state spaces with discrete time space is of great importance. The authors have made the effort to push further the existing methods for the current setting.

**Weaknesses:**

The notation is confusing and important details are often omitted.  The assumptions seem to be rather stringent. The paper lacks mathematical clarity. See the below sections for details.

### Mathematical comments

- *Proposition 1*. The formula for $q_t$ depends on the initial data distribution $p_{data}$. When $t \rightarrow \infty$, the term on the right side  $\frac{1}{S} \left(1 - e^{-t}\right) \mathbf{1}_S \mathbf{1}_S^\top + e^{-t} I_S \rightarrow \frac{1}{S} \mathbf{1}_S \mathbf{1}_S^\top$ . The latter is a fixed matrix and thus $q_t \rightarrow \frac{1}{S} \mathbf{1}_S \mathbf{1}_S^\top \cdot p_{data}$. This is not necessarily the uniform distribution.
- *Proposition 1* What does approaching mean here? In what sense do you have the convergence?
-  *line 280* . The sentence `With rate $Q^{\leftarrow}_t$, it holds that...` is not clear. Is there a reference or proof for this statement?
- *Assumption 2* It does not seem to be easy to verify this assumption. Is there a general class of distributions that are known to satisfy it?
- *Theorem 2* The number $\kappa_i$ is not always well-defined, as we deal with discrete distributions, where some of the probabilities may be equal to zero.

#### typos
- *line 187* if setting β(t) as a time-dependent scala
- *line 206* Nota -> Note
- *line 323* For completeness, We -> For completeness, we

**Questions:**

- *Equation on line 723*. Why is $Q$ equal to the Kronecker sum $Q^{tok}$?
 - *line 296* Why is the reverse process time-inhomogeneous?
 - *line 3 of the Algorithm*: How easy is it to find the maximum of the estimated score function? Is there any concavity assumption to make this problem solvable in theory/real-time? Is this assumption satisfied for practically relevant examples?
 - *line 361* Why is there a uniform upper bound on all the scores and their estimators? On line 371 the authors mention that the score function may be as large as infinity for some data points. This means, that when $\delta$ is small, this uniform upper bound $C_1$ becomes larger, thus, it is dependent on $\delta$. Can this dependence be quantified?
 - *Equation (4)*. Why is this true? Is there a reference or a proof for this statement?

---

> ### Author Response · Authors · 2024-11-20
> **Part 1**
>
> We appreciate your detailed review. Our responses are provided below, and all revisions have been highlighted in blue in the revised manuscript.
>
> ---
>
> **Q1**.   Proposition 1. The formula for $q_t$ depends on the initial data distribution $p_\mathrm{data}$. When $t\rightarrow \infty$, the term on the right side $\frac{1}{S}(1-e^{-t})\mathbf{1}_S \mathbf{1}_S^\top+e^{-t} I_S\to \frac{1}{S}\mathbf{1}_S\mathbf{1}_S^\top$. The latter is a fixed matrix and thus $q_t \rightarrow \frac{1}{S}\mathbf{1}_S\mathbf{1}_S^\top \cdot p\_\mathrm{data}$. This is not necessarily uniform distribution.
>
> **A1**.  This is a misunderstanding and we would like to clarify this mathematical detail here. When $t \to +\infty$, $[\frac{1}{S}(1-e^{-t})\mathbf{1}_S \mathbf{1}_S^\top+e^{-t} I_S]^{\otimes d}\to [\frac{1}{S}\mathbf{1}_S \mathbf{1}_S^\top]^{\otimes d} =\frac{1}{S^d}\mathbf{1}\_{S^d}\mathbf{1}\_{S^d}^\top$. This yields that $q_t = [\frac{1}{S}(1-e^{-t})\mathbf{1}\_S \mathbf{1}\_S^\top+e^{-t} I_S]^{\otimes d} \cdot p\_\mathrm{data} \to \frac{1}{S^d} \mathbf{1}\_{S^d} \mathbf{1}\_{S^d}^\top \cdot p\_\mathrm{data} = \frac{1}{S^d} \mathbf{1}\_{S^d} \cdot [\mathbf{1}\_{S^d}^\top \cdot p\_\mathrm{data}] = \frac{1}{S^d} \mathbf{1}\_{S^d} = \pi^d (t \to +\infty)$, where $\pi^d$  is the uniform distribution on $[S]^d$ and the equality $\mathbf{1}\_{S^d}^\top\cdot p\_\mathrm{data}=1$ holds because $p\_\mathrm{data}$ is a discrete probability distribution whose entries sum to 1. This means that $q_t \to \pi^d (t \to +\infty)$.
>
> We have added this clarification to the revised version of our paper on Line 755.
>
> ---
>
> **Q2**. Proposition 1 What does approaching mean here? In what sense do you have convergence?
>
> **A2**.  The marginal distribution $q_t\in\mathbb{R}^{|\mathcal{X}|}$ has each component defined as a function of $t$. Here, "approaching" refers to taking the limit as $t\to +\infty$. As $t \to +\infty$, the marginal distribution $q_t$ converges, in the sense of a function limit, to $\pi^d=\frac{1}{S^d}\mathbf{1}\_{S^d}$ , the uniform distribution over $\mathcal{X}=[S]^d$.
>
> ---
>
> **Q3**. line 280 . The sentence 'With rate $Q^{\leftarrow}_t$, it holds that...' is not clear. Is there a reference or proof for this statement?
>
> **A3**. This result directly follows from Campbell et al. (2022, Proposition 3), as cited in Line 269. Proposition 3 in Campbell et al. (2022) explicitly provides the expression for the reverse rate matrix corresponding to a given forward rate matrix. The CTMC with this reverse rate is the exact time reversal of the forward CTMC. For clarity, we have revised the statement in our paper on Lines 269-274:
> According to Campbell et al. (2022), the reverse process $Y=(Y_t)\_{t\in [0,T]}$ can be achieved by a CTMC starting from $Y_0\sim q_T$ with $\mathrm{Law}(Y_t)\stackrel{\text{a.s.}}{=}\mathrm{Law}(X_{T-t})=q_{T-t}$, and the reverse rate $Q^{\leftarrow}\_t\in\mathbb{R}^{S^d\times S^d}$ is of the form
> $Q^{\leftarrow}\_t(\mathbf{x},\tilde{\mathbf{x}})=\sum_{i=1}^d Q_{T-t}^\mathrm{tok}(\tilde{\mathbf{x}}^i,\mathbf{x}^i)\delta \\{\mathbf{x}^{\backslash i},\tilde{\mathbf{x}}^{\backslash i} \\} \frac{q_{T-t}(\tilde{\mathbf{x}})}{q_{T-t}(\mathbf{x})}.$
>
> ---
>
> **Q4**. Assumption 2 It does not seem to be easy to verify this assumption. Is there a general class of distributions that are known to satisfy it?
>
> **A4**. Assumption 2 is satisfied by a broad class of practical distributions. A particularly important example is when the data distribution consists of independent and identically distributed (i.i.d.) components, where each marginal distribution has full support on $[S]$. In this case,  $p\_\mathrm{data}$ has the form of Kronecker product decomposition as $p\_\mathrm{data}=p\_\mathrm{data}^1\otimes p\_\mathrm{data}^2\otimes\cdots\otimes p\_\mathrm{data}^d$ where $p\_\mathrm{data}^i\in\mathbb{R}^S$ is the marginal distribution of the $i$-th token of the data on the state space $[S]$ and $p\_\mathrm{data}^1=\cdots=p\_\mathrm{data}^d.$ By independence, we have that $s_0(\mathbf{x})\_{i,\hat{x}^i}=\frac{p\_\mathrm{data}(\mathbf{x}^{\backslash i}\odot \hat{x}^i)}{p\_\mathrm{data}(\mathbf{x})}=\frac{p\_\mathrm{data}^i(\hat{x}^i)\prod_{j\neq i}p\_\mathrm{data}^j(x^j)}{\prod_{j=1}^d p\_\mathrm{data}^j(x^j)}=\frac{p\_\mathrm{data}^i(\hat{x}^i)}{p\_\mathrm{data}^i(x^i)}$ （$i\in [d]$, $\mathbf{x}\in [S]^d$, and $\hat{x}^i\neq x^i$）, which is exactly the score function of the marginal distribution $p\_\mathrm{data}^i$. Thus, we know that the uniform upper bound $L$ in Assumption 2 is exactly the uniform upper bound of the score of $p\_\mathrm{data}^i$, which is only dependent on $S$  but not on $d$.
>
> We have added this supplementary explanation for Assumption 2 to the paper on Lines 378-379.

---

> ### Author Response · Authors · 2024-11-20
> **Part 2**
>
> ---
>
> **Q5**. Theorem 2 The number $\kappa_i$ is not always well-defined, as we deal with discrete distributions, where some of the probabilities may be equal to zero.
>
> **A5**.  We would like to clarify this misunderstanding. Theorem 2 holds under Assumption 2 which ensures that the data distribution $p\_\mathrm{data}$  is strictly positive. The strict positiveness of $p\_\mathrm{data}$ naturally leads to the strict positiveness of marginal distribution $p\_\mathrm{data}^i$ as $p\_\mathrm{data}^i(x)=\sum\_{\mathbf{x}\in [S]^d:\ \mathbf{x}^i=x}p\_\mathrm{data}(\mathbf{x})>0$ for any $x\in [S]$ and $i\in [d]$. Thus, $\kappa_i=\frac{(p\_\mathrm{data}^i)\_\mathrm{max}}{(p\_\mathrm{data}^i)\_\mathrm{min}}$ is well-defined and would not blow up or equal to zero. We have clarified this point on Lines 392-393 in our revised paper.
>
> ---
>
> **Q6**. Typos
>
> **A6**.  We appreciate the reviewer for identifying these typos. We have corrected them accordingly on Lines 211 and 322.
>
> ---
>
> **Q7**. Equation on line 723. Why is $Q$ equal to the Kronecker sum $Q^\mathrm{tok}$?
>
> **A7**. The Kronecker product structure of $Q$ was inspired by Chen and Ying (2024, Proposition 3) where $d=2$ is considered.  This structure can be verified by direct calculation from the definition of $Q$ and $Q^\mathrm{tok}$:
>
> First, the transition rate matrix for each dimension $Q^\mathrm{tok}=\frac{1}{S}(\mathbf{1}_S\mathbf{1}_S^\top-S\cdot I_S)$ satisfies that $Q^\mathrm{tok}(x,y)=\frac{1}{S}$ for $x\neq y\in [S]$ and $Q^\mathrm{tok}(x,x)=\frac{1}{S}-1$ for $x\in [S]$.
>
> We next calculate the Kronecker sum $\bigoplus_{k=1}^d Q^\mathrm{tok}=\sum_{k=1}^d I_S^{\otimes (k-1)}\otimes Q^\mathrm{tok}\otimes I_S^{\otimes (d-k)}$. By the definition of the Kronecker product, we know that for the Kronecker product of $d$ square matrices $A_1, \cdots, A_d\in \mathbb{R}^{S\times S}$, its elements can be expressed using the following indexing relationship: $(A_1 \otimes A_2 \otimes \cdots \otimes A_d)(i_1 i_2 \cdots i_d, j_1 j_2 \cdots j_d) = A_1(i_1, j_1)\cdot A_2(i_2, j_2) \cdots A_d(i_d, j_d)$, where $i_1 i_2 \cdots i_d $ and $j_1 j_2 \cdots j_d$ are multi-dimensional indices representing the specific row and column positions in the $S^d\times S^d$ space. Thus, by the structure of $Q^\text{tok}$ and the identity matrix $I_S$, we can see that for any $\mathbf{x}\in [S]^d$, $i\in [d]$, and $\hat{x}^i\neq x^i$, it holds that $\left[\bigoplus_{k=1}^d Q^\mathrm{tok}\right](\mathbf{x},\mathbf{x}^{\backslash i}\odot \hat{x}^i)=\sum_{k=1}^d \left[I_S^{\otimes (k-1)}\otimes Q^\mathrm{tok}\otimes I_S^{\otimes (d-k)}\right](\mathbf{x},\mathbf{x}^{\backslash i}\odot \hat{x}^i)=\left[I_S^{\otimes (i-1)}\otimes Q^\text{tok}\otimes I_S^{\otimes (d-i)}\right](\mathbf{x},\mathbf{x}^{\backslash i}\odot \hat{x}^i)=\prod_{j\neq i}I_S(x^j,x^j) \cdot Q^\text{tok}(x^i,\hat{x}^i)=1^{d-1}\cdot \frac{1}{S}=\frac{1}{S}.$
>
> Similarly, we have that for any $\mathbf{x}\in [S]^d$, $\left[\bigoplus_{k=1}^d Q^\mathrm{tok}\right] (\mathbf{x}, \mathbf{x})=\sum_{k=1}^d \left[I_S^{\otimes (k-1)}\otimes Q^\mathrm{tok}\otimes I_S^{\otimes (d-k)}\right] (\mathbf{x},\mathbf{x})=\sum_{k=1}^d (\frac{1}{S}-1)=(\frac{1}{S}-1)d.$
>
> Hence, we conclude that $Q = \bigoplus\_{k=1}^d Q^\mathrm{tok}=\sum\_{k=1}^d I_S^{\otimes (k-1)}\otimes Q^\mathrm{tok}\otimes I_S^{\otimes (d-k)}.$
>
> We have added these mathematical details in our revised paper on Lines 720-745.
>
> ---
>
> **Q8**. line 296 Why is the reverse process time-inhomogeneous?
>
> **A8**. The reverse rate matrix $Q_t^{\leftarrow}$ is time-dependent, which indicates that the reverse process is time-inhomogeneous, according to the definition of a time-inhomogeneous CTMC (see, e.g., Holmes-Cerfon (2022)).

---

> ### Author Response · Authors · 2024-11-20
> **Part 3**
>
> **Q9**.  line 3 of the Algorithm: How easy is it to find the maximum of the estimated score function? Is there any concavity assumption to make this problem solvable in theory/real-time? Is this assumption satisfied for practically relevant examples?
>
> **A9**. We appreciate the reviewer's insightful observation. In Line 3 of Algorithm 1, we have set $\lambda_k=\max\_{\mathbf{x} \in \mathcal{X}} \\{ \hat{s}\_{T-kh}(\mathbf{x})\_\mathbf{x}\\}$, where the maximum is obtained over the discrete set $\mathcal{X}=[S]^d$. When $S^d$ is relatively small, taking the maximum is practically reasonable. When $S^d$ is large, it may be impractical to get the exact maximum. By the uniformization of CTMC (Chen and Ying, 2024), we can instead get an upper bound for $\max_{\mathbf{x} \in \mathcal{X}} \\{ \hat{s}\_{T-kh}(\mathbf{x})\_\mathbf{x} \\}$ to implement Algorithm 1.  By applying score clipping as discussed in Appendix A.4, we can ensure that $\hat{s}\_{T-kh}(\mathbf{x})\_\mathbf{x} \leq \frac{3}{2}dS\left(1+\frac{S}{e^{T-kh}-1}\right)$ for all $\mathbf{x}\in\mathcal{X}$. Therefore, we can set $\lambda_k = \frac{3}{2}dS\left(1+\frac{S}{e^{T-kh}-1}\right)$ which is tractable given $k, T, h, d, S$ , and thus circumvent obtaining the maximum over the set $\mathcal{X}$ .
>
> In the revised version of our paper, we have added the discussion on $\lambda_k$ on Lines 1330-1338 to clarify this point.
>
> ---
>
> **Q10**. Line 361 Why is there a uniform upper bound on all the scores and their estimators?
>
> **A10**.  The fact that $C_1$ and $C_2$ are the uniform upper bounds for all the scores and their estimators has been deferred to the Appendix. By Lemmas 2 and 4 in Appendix A, we know that $C_1$  and $C_2$ are uniform bounds for all the scores and their estimators with and without early stopping respectively ($C_1$ and $C_2$ are defined in Theorem 1 and Theorem 2 respectively). See also discussions on $C_1$ and $C_2$ in Appendix A.4 (Lines 961-971 in our paper).
>
> We have added the explanation for this statement on Line 362 of the revised paper.
>
> ---
>
> **Q11**. On line 371 the authors mention that the score function may be as large as infinity for some data points. This means, that when $\delta$ is small, this uniform upper bound $C_1$ becomes larger, thus, it is dependent on $\delta$. Can this dependence be quantified?
>
> **A11**. Yes, the dependence on $\delta$ of $C_1$ is quantified in Theorem 1 (Line 349): $C_1=\max\left\\{1+\frac{S}{\delta},\max\_{\mathbf{x}\in\mathcal{X}, k\in\\{0,\cdots,K-1\\}}\Vert\hat{s}\_{(k+1)h+\delta}(\mathbf{x})\Vert_\infty\right\\}$ . By applying score clipping, we can ensure that $C_1\leq \frac{3}{2}\left(1+\frac{S}{\delta}\right)$ (see discussions on $C_1$ in Appendix A.4 (Lines 961-971).
>
> ---
>
> **Q12**. Equation (4). Why is this true? Is there a reference or a proof for this statement?
>
> **A12**. By Campbell et al. (2022, Proposition 3),  $q_{T-t}$ is the marginal along the reverse process $(Y_t)\_{t\in [0,T]}$: $\mathrm{Law}(Y_t)\stackrel{\text{a.s.}}{=}\mathrm{Law}(X_{T-t})=q_{T-t}$. With the rate matrix $Q_t^\leftarrow$ of the reverse CTMC, we can write out the marginal $q_{T-t}$ satisfies the Kolmogorov equation (see, e.g., Campbell et al. (2022); Chewi (2023), as cited in Line 186 of the revised paper): $\frac{\mathrm{d} q_{T-t}}{\mathrm{d} t} = Q^{\leftarrow}\_t{}^\top q_{T-t}.$
>
> ---
>
> **References**
>
> [1] Andrew Campbell, Joe Benton, Valentin De Bortoli, Thomas Rainforth, George Deligiannidis, and Arnaud Doucet. A continuous time framework for discrete denoising models. Advances in Neural Information Processing Systems, 35:28266–28279, 2022.
>
> [2] Hongrui Chen and Lexing Ying. Convergence analysis of discrete diffusion model: Exact implementation through uniformization. arXiv preprint arXiv:2402.08095, 2024.
>
> [3] Sinho Chewi. Log-concave sampling, 2023. Book draft available at https://chewisinho.github.io.
>
> [4] Miranda Holmes-Cerfon. Lecture 4: Continuous-time Markov Chains, 2022. URL https://personal.math.ubc.ca/~holmescerfon/teaching/asa22/handout-Lecture4_2022.pdf Lecture notes, University of British Columbia.

---

> > ### Comment · Reviewer_Gwu3 · 2024-11-27
> >
> > My concerns were properly addressed. I raised my score.

---

> > > ### Author Response · Authors · 2024-11-27
> > > **Thank you**
> > >
> > > Thank you for raising your score. We appreciate your detailed review and are glad that our rebuttal has addressed your concerns.

---

> ### Author Response · Authors · 2024-11-22
>
> Dear Reviewer Gwu3,
>
> Thank you for your detailed review. We have carefully addressed each point in our response and revised the manuscript accordingly. We would be grateful for any additional feedback or questions you may have.
>
> Best,
>
> Authors

---

> ### Author Response · Authors · 2024-11-25
> **We Are Looking Forward to Your Reply**
>
> Dear Reviewer Gwu3,
>
> Thank you for your detailed review. We have addressed your concerns point by point in our rebuttal and made corresponding revisions. Given the approaching PDF revision deadline, your feedback would be greatly appreciated, particularly if our explanation of the distribution convergence results in Proposition 1 addresses your concerns. We would be happy to address any additional questions you may have.
>
> Best regards,
>
> Authors

---

### Meta-Review · Area_Chair_EqKk · 2024-12-13

**Metareview:**

This paper offers a convergence analysis of score-based discrete diffusion model when the state space is $[S]^d$, generalizing the prior work of Chen and Ying that handles the case of $\{0,1\}^d$. The work leverages heavily the techniques from Chen and Ying, but also offers new ingredients to handle the generalization. Given the importance of discrete diffusion, the work provides a useful contribution to the theoretical understanding of such models.

**Additional Comments On Reviewer Discussion:**

The rebuttal has helped clarify the contribution of this work in relation to existing works, as well as some technical details, leading to a consensus score of acceptance.

---

### Decision · Program_Chairs · 2025-01-22

Accept (Poster)